# Sniffer worm, *C. elegans*, as a toxicity evaluation model organism with sensing and locomotion abilities

**Jun Sung Kim[1], Sang-Kyu Park[2], Haeshin Lee**[1]*

**1** Department of Chemistry KAIST, Daejeon, Republic of Korea, **2** Department of Medical Biotechnology, College of Medical Science, Soonchunhyang University, Asan, Chungnam, Korea

* haeshin@kaist.ac.kr

**Data Availability Statement:** All relevant data are within the manuscript and its Supporting information files.

**Funding:** This work was supported by a National Research Foundation of Korea (NRF) grant funded

## Abstract

Additive manufacturing, or 3D printing, has revolutionized the way we create objects. However, its layer-by-layer process may lead to an increased incidence of local defects compared to traditional casting-based methods. Factors such as light intensity, depth of light penetration, component inhomogeneity, and fluctuations in nozzle temperature all contribute to defect formations. These defective regions can become sources of toxic component leakage, but pinpointing their locations in 3D printed materials remains a challenge. Traditional toxicological assessments rely on the extraction and subsequent exposure of living organisms to these harmful agents, thus only offering a passive detection approach. Therefore, the development of an active system to both identify and locate sources of toxicity is essential in the realm of 3D printing technologies. Herein, we introduce the use of the nematode model organism, *Caenorhabditis elegans* (*C. elegans*), for toxicity evaluation. *C. elegans* exhibits distinctive 'sensing' and 'locomotion' capabilities that enable it to actively navigate toward safe zones while steering clear of hazardous areas. This active behavior sets *C. elegans* apart from other aquatic and animal models, making it an exceptional choice for immediate and precise identification and localization of toxicity sources in 3D printed materials.

## Introduction

A critical and indispensable step in developing new materials for biomedical applications is the toxicological evaluation of novel materials [1]. Assessment of material toxicity fundamentally requires testing on living organisms. Early toxicological studies employed plant species, such as *Lemna gibba* [2]. *Lemna gibba*'s small size, simple cultivation process, and short life cycle render it a practical candidate for toxicity assessments [3, 4]. The primary advantage of utilizing this plant model resides in the observation of growth inhibition induced by toxic elements. However, the outcomes obtained from *Lemna gibba* growth inhibition studies do not directly parallel the toxicological impacts on humans.

Due to the discrepancies between plant-based toxicity models and human toxicity outcomes, subsequent research has embraced the use of animal models. These models can be

by the Korea government (MSIT) (No. 2018R1A5A1025208) and the Disaster and Safety Management Institute (MPSS-CG-2016-02) from The Ministry of Public Safety and Security of the Republic of Korea (JSK). The funders had no role in study design, data collection and analysis, decision to publish, or preparation of the manuscript.

**Competing interests:** The authors have declared that no competing interests exist.

broadly categorized into two groups. One involves aquatic organisms, such as *Daphnia* [5, 6] or *Danio rerio* (zebrafish) [7, 8] and the other is terrestrial animals, including rats or mice [9]. Toxicological effects observed in aquatic animals are assessed by various factors, such as acute immobilization, mobility, survival, developmental disorder, and/or embryo hatching [6]. However, the findings from aquatic settings vary significantly from those observed in terrestrial conditions, thus inclining researchers towards the use of terrestrial animal models. Rats and mice have been dominantly chosen as animal models due to small size, affordability, and high reproductive rates. Nonetheless, the recent enforcement of rigorous and comprehensive ethical regulations has introduced considerable challenges to the employment of these model systems [10]. Consequently, organ-on-a-chip technologies have been introduced as alternative approaches [11]. Additionally, cosmetic science has started to find new methods for assessing the toxicity of novel components.

The challenges associated with current model systems have motivated researchers to develop convenient and ethically sound methods for toxicity evaluation. Importantly, additive manufacturing technologies known as 3D printing also necessitate the development of new toxicity assessment systems [12]. Compared with objects manufactured through a conventional casting-based method, ones produced by 3D printing inherently exhibit increased defective regions. This is primarily due to the intrinsic layer-by-layer deposition processes in additive manufacturing, which inherently involve multiple iterations, during which the generation of defects is inescapable. In stereolithography (SLA) methods, variances in factors such as light intensity, penetration depth, and the concentration and dispersion inhomogeneity of polymers can arise [13]. Furthermore, UV-light irradiation triggers a homolytic bond cleavage, resulting in the creation of highly reactive radical species, which serve as the roots of chemical toxicity. In fused deposition modeling (FDM), fluctuations in nozzle temperature significantly contribute to the development of local defects during the printing process [14]. Given that 3D printing techniques are optimally suited for small-scale prototype production, each product may exhibit distinctive defect sites, thereby leading to disparate levels of toxicity.

Consequently, it becomes imperative to develop a novel animal model capable of detecting variations in toxicity sources between individual products. Unfortunately, the existing plant and animal models are insufficient for this purpose (Table 1). Plant models, lacking locomotive capabilities, fail to discern toxicological sources within printed materials. Although zebrafish, an aquatic model used in toxicity evaluations, possess locomotive abilities, the diffusion of leached toxic compounds in water impedes their effectiveness in accurately locating toxicity sources within printed materials. Additionally, foreign body reactions to implanted materials in mice and rats further complicate identifying toxicity sources.

**Table 1. Typical model organism list and their chemical detection ability and locomotion.**

| | Model system | Sensing | Locomotion | Remarks |
|---|---|---|---|---|
| Animal | Zebrafish | O | O | Lives underwater |
| | *Daphnia sp.* | O | O | |
| | Rodent(rat, mouse) | O | O | Ethical issues |
| | *C. elegans* | O | O | |
| Plant | *Lemna gibba* | O | X | Does not move |
| | *Arabidopsis thaliana* | O | X | |
| Others | Mammalian cell lines | O | X | Cannot move in a reasonable time |
| | Fungi(*Saccharomyces cerevisiae*) | O | X | |
| | Bacteria(*Escherichia coli*) | O | X | |

Given these challenges, it becomes critical to introduce a toxicity model that unifies both sensing and locomotion capabilities. We propose the employment of *Caenorhabditis elegans* (*C. elegans*), a model endowed with the capacity to discern both the overall toxicity and specific toxic sites within printed materials. *C. elegans* was chosen due to its inherent chemotactic behavior [15]. The crawling activity of the nematode serves as phenotypic indicators pivotal for pinpointing the sources of toxicity within test materials. Furthermore, the lifespan of *C. elegans* is short from two to four weeks, making it a suitable model system [16]. As one of the few organisms with a fully sequenced genome [17], the adult worm measures approximately 1 mm in length, with well-documented differentiation pathways for all 959 cells originating from a single germinal cell [18]. This feature makes *C. elegans* an ideal candidate for evaluating toxicity effects. This attractive animal has already been studied in various fields, including aging, new drug development, and neurobiology [19–23]. Yet, the potential of *C. elegans* as an 'active' toxicological model, integrating both sensing and locomotion capabilities, has been unexplored.

## Materials and methods

### Materials

A stereo-lithographic 3D printer (Master Plus J) and its resin (Arario 410) containing diglycidyl ether of bisphenol (20–40%), acryloyl morpholine (10–20%), tripropylene glycol diacrylate (20–40%), 1-hydroxycyclohexyl phenyl ketone (3–7%), diphenylphosphine oxide (1–3%), and 2-methyl-4-2-morpholinopropiophenone (1–3%) were purchased from Carima CO, Korea. A dissecting microscope was purchased from Olympus (SZ61, Tokyo, Japan). The 3D-printed blocks in this study were designed using the commercial 3D modeling program Rhino 3D (Rhinoceros version 5 SR 12, USA). Except those reported above, all chemical agents were purchased from Sigma-Aldrich CO. St. Louis, MO, USA.

### Preparation of 3D-printed blocks, extracts and food sources

3D-printed products were printed in dimensions of 10 mm by 10 mm with a height of 1 mm. The 80 printed blocks and magnetic stirrer were added to 400 mL of Luria-Bertani broth (LB) media, and extraction was carried out for 72 hours at room temperature. During the extraction, the bottle was protected from light. Then, the mixture was filtered by a bottle top filter (Sartolab® BT vacuum filter 180C5E, pore size of 0.22 μm). After all the steps mentioned above, Escherichia coli OP50 (uracil auxotroph) was grown in 20 ml of extracted LB media at 37°C with shaking (220 rpm) for 16 hours.

### *C. elegans* maintenance

The N2 CGCb strain was used as a wild-type control. The CL2070 (hsp-16.2::gfp) and CF1553 (sod-3::gfp) strains were kindly given to us by Sang-Kyu Park (Department of Medical Biotechnology, SoonChunHyang Univ.). Nematode growth medium (NGM) containing 25 mM NaCl, 1.7% agar, 2.5 mg/ml peptone, 5 μg/mL cholesterol, 1 mM $CaCl_2$, 1 mM $MgSO_4$, and 50 mM $KH_2PO_4$ (pH 6.0) was used as a growth medium. All experiments were conducted at 20 °C unless otherwise mentioned. *E. coli* OP50 was added to each NGM plate as a food source.

### Lifespan assay

Sixty age-synchronized 3-day-old worms were transferred to fresh NGM plates, and 5-fluoro-2'-deoxyruridine (FUdR) was added to prevent internal hatching. Thereafter, worms were transferred to fresh NGM plates with FUdR every other day until all worms were dead. The

number of living and dead worms was scored every day. Three independent replicate experiments were performed. Statistical analysis was performed using the log-rank test [24].

### GFP expression of stress-response genes

Age-synchronized CL2070 (hsp-16.2::gfp) and CF1553 (sod-3::gfp) worms were placed in NGM plates seeded with contaminated OP50 at 20 ˚C for 5 days. Then, the worms were mounted on a slide glass coated with 2% agarose and anaesthetized with 1 M sodium azide. After covering the slide with a coverslip, the expression of each reporter gene was observed using a confocal microscope (Nikon FV10i, Nikon, Tokyo, Japan). The fluorescence intensity of a randomly selected single worm was quantified with a fluorescence multiplate reader (Synergy MX, BioTek, Winooski, America).

### Fertility assay

For the fertility assay, five late L4 or early young adult stage worms were transferred to a fresh NGM plates seeded with contaminated OP50 and permitted to lay eggs for 5 hrs. The eggs were maintained at 20 ˚C for 2 days. A single worm was transferred to a fresh NGM plates seeded with contaminated OP50 every day until it laid no eggs. Eggs spawned by a single worm were incubated at 20 ˚C for 48 hours, and the number of progeny produced was recorded each day. The average number of progeny produced by 5 worms treated with 3D-printed block extract was compared to that produced by control worms.

### Body bending assay

This experiment was performed with a slight modification of the previously known *C. elegans* body bending assay [21]. First, this experiment was carried out by using the result of extraction of 3D-printed blocks with double distilled water (DDW) instead of LB. The 8 3D-printed blocks and magnetic stirrers were added to 40 mL of DDW, and extraction was carried out for 72 hours at room temperature (Fig 2C). During the extraction, the bottle was protected from light. Then, the mixture was filtered by a syringe filter (Satorius Minisart® syringe filter, pore size of 0.2 μm). Upon completion of the extraction, 100 μl of NGM was spread evenly, and OP50 was seeded. After that, we transferred a 3-day-old nematode (i.e., the 3rd day from egg laying, corresponding to L4/young adult stage) to the prepared NGM. Before the start of this assay, 5-day-old worms (i.e., the 5th day from egg laying, corresponding to adult stage) were put on a sterile fresh NGM agar plate without OP50 and allowed to crawl freely to remove the agglomerated bacteria from them. After visually examining whether the bacteria were removed, a worm was put into a 96-well plate (SPL 30096, SPL life science CO., Korea) with M9 buffer and allowed to swim freely for 1 min for acclimatization to the environment. Then, the number of thrashes was counted for 1 min. We monitored at least 6 worms for each assay. A worm movement that swings its head and/or tail to the same side is counted as one thrash.

### 3D-printed block extract analysis using X-ray photoelectron spectroscopy

The 3D-printed block was extracted with DDW, and 10 μl of the resultant extracted was dropped on a 10 mm by 10 mm Si wafer. Then, the DDW was evaporated naturally. X-ray photoelectron spectroscopy was performed with a Thermo Scientific™ K-Alpha™ X-ray photoelectron spectrometer (XPS) system.

### Dead-zone detection assay

The 3D-printed block was further cured using a piece of aluminum foil, as shown in Fig 4A, as a photomask covering only half of the block, followed by additional UV irradiation. After that, we put the produced block in the center of the testing plate and put *C. elegans* worms at the same age of 3-day-old (i.e., the 3rd day from egg laying, corresponding to L4/young adult stage) in four places 20 mm away from the center (Fig 4C). After incubation at room temperature for 1 hr, we counted the worms that were found within 3 mm, 6 mm and 10 mm from the surface of the block. The following equation was used to quantify the distribution of worms (ⓐ& ⓑ are shown in Fig 4C).

$$\% \text{ of worms} = \frac{(\# \text{ of worms in } ⓐ \text{ or } ⓑ)}{(\# \text{ of worms in } ⓐ + \text{ of worms in } ⓑ)} \times 100$$

## Results and discussion

### Toxic substance effects on *C. elegans*' lifespan

Release of toxic compounds released from the 3D-printed cuboids was monitored by observing the lifespan of *C.elegans*. Fig 1B shows the differences in the maximal lifespan of *C. elegans*. When *C.elegans* was exposed to the toxic compounds through the *Escherichia. Coli* (OP50) supplemented with toxic compounds extracted from the cuboids in Luria-Bertani broth (LB) media, the lifespan was reduced from 24 (black) to 20 (red) days. We hypothesized that toxic compounds released from the 3D-printed cuboids are absorbed by OP50 during the growth in LB media. A control experiment was carried out with the OP50 grown in LB media without the toxic compounds extracted from the cuboids. The mean lifespan, defined by 50% survival of *C. elegans*, was 15 days for the control, which was decreased to 13 days for the treatment group (dotted lines, Fig 1B). Three independent experiments were performed, and all detailed lifespan parameters are listed in Table 2. The decrease in longevity caused by the toxic compounds released from the 3D-printed cuboid was 15.8% (p<0.05). FUdR has been known to affect the differential expression of various genes. However, negligible effects of FUdR on the lifespan of wild-type has been reported [25–28].

### Relative expression levels of a stress-response gene

The lifespan reduction of *C. elegans* exposed to the extract of a 3D-printed cuboid indicates that there might be a gene indicator to provide further evidences at a molecular level. We selected *hsp-16.2* and *sod-3*. *Hsp-16.2*, a stress-responsive reporter gene, has been known to predict longevity in *C. elegans*, and its expression correlates with increased resistance to heat-shock stress and enhanced lifespan [22, 29]. *Sod-3*, on the other hand, plays a role in the cellular enzymatic defense system against reactive oxygen species [30]. When the worms were exposed to the 3D-printed cuboid extracts for 2 days post-adulthood, we observed a significant downregulation of *hsp-16.2* in the worms treated with 3D-printed cuboid extract (Fig 1C upper panels and 1D left bars). The observation was made with the green fluorescent images due to the expression of the fusion protein HSP-16.2-GFP (Fig 1C). The qualitative analysis showed a considerable decrease in overall green fluorescence. In addition, quantitative analysis (n = 20 for each group) was also performed. The relative expression level of hsp-16.2 decreased to 52.96 ± 6.37% for the group supplemented with the 3D-printed cuboid extracts (Fig 1D, left bars). A comparable result was obtained with SOD-3-GFP expression (74.92 ± 6.34% for the 3D cuboid extract group) (Fig 1C lower panels and 1D right bars). These findings suggest that

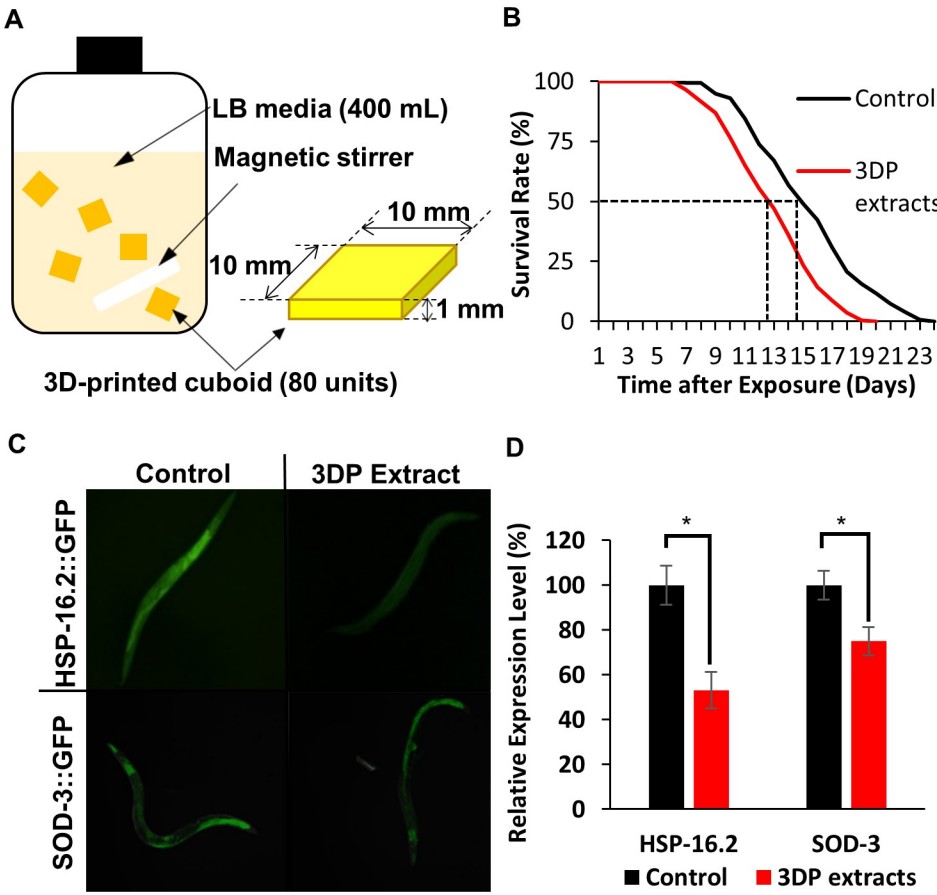

**Fig 1. Procedure of 3D-printed cuboids in LB and the effect of extracts on *C. elegans* health.** (A) Schematic description of extracting toxic compounds from the 3D-printed cuboids. (B) Lifespan reduction of *C. elegans* by 3D-printed cuboid extracts (red line). Dotted lines indicate 50% survival of *C. elegans*. (C) Stress-response gene expression levels: HSP-16.2::GFP (upper panels) SOD-3::GFP (lower panels) in response to LB extract with or without released toxic compounds. (D) Quantitative analysis of the stress-responsive protein expression levels shown in Fig 1C. The black bars are the results obtained using uncontaminated LB supplementation, and the red bars are the results obtained using the LB contaminated with released toxic compounds. The asterisks indicate p-values less than 0.05 compared with the control.

the nematodes exposed to 3D-printed cuboid extracts have diminished the stress resistance capabilities, supporting our previous findings regarding the reduction in lifespan.

## Toxic substance effects on *C. elegans*' fertility

We thought that the toxic substances released from the cuboids could impact the fertility of *C. elegans*. Normally, healthy nematode can hatch approximately 300 eggs throughout its lifetime [31], resulting in the presence of tens of eggs even after one day of culturing a single *C. elegans* organism [32]. Exposure to the toxic substances in LB medium can adversely affect the fertility of *C. elegans*. Thus, we designed an experiment to investigate the influence of these toxic effects on their fertility. Firstly, all *C. elegans* nematodes were synchronized in their age. Second, a single worm was transferred to one plate of nematode growth medium (NGM). Third, incubation at 20 ˚C was carried out for 24 hrs, and then the incubated *C. elegans* was transferred again to a new NGM plate. The empty plate incubated for 24 hrs (Fig 2A) was not discarded but rather

**Table 2. Reduction in the lifespan of *C. elegans* by 3D-printed block extract.**

|  |  | Mean Lifespan (days) [a] | Max. Lifespan (days) [b] | *p*-value [c] | % decreased [d] |
|---|---|---|---|---|---|
| 1st | Control | 16.68 | 24 | <0.001 | 20.83 |
|  | Extracts | 13.20 | 20 |  |  |
| 2nd | Control | 14.95 | 23 | <0.05 | 13.50 |
|  | Extracts | 12.93 | 19 |  |  |
| 3rd | Control | 14.88 | 22 | <0.05 | 12.48 |
|  | Extracts | 13.02 | 19 |  |  |
| Average | Control | 15.50 | 23 | <0.05 | 15.80 |
|  | Extracts | 13.05 | 19.33 |  |  |

[a] Mean lifespan is the day when 50% of worms survived.

[b] Max. lifespan is the greatest age reached by the last surviving worm.

[c] p-value was calculated using the log-rank test by comparing the survival of the control group and the group exposed to 3D-printed block extract (marked as "Extracts" in the table).

[d] % effects were calculated by (C-E)/C*100, where E is the mean lifespan of *C. elegans* treated with extracts and C is the mean lifespan of the control *C. elegans*.

placed in the same incubator (20 ˚C) to observe the *C. elegans* offspring hatched from the remaining eggs (# marked plates in Fig 2A). The transfer process of the single *C. elegans* worm was continued for 6 days until no hatched *C. elegans* offspring were found. We checked the number of offspring at two days after each transfer. Fig 2B shows the mean number of progeny hatched by 5 individual worms on each day. The overall trend regarding the number of offspring per day showed a decreased number of offspring from the *C. elegans* exposed to the toxic compound LB extract: 8 at 2 days, 34 at 3 days, 11 at 4 days, and 4 at 5 days. Asterisks indicate p-values less than 0.05 compared with the control. Assessing the total progeny produced by the worms is a more appropriate measure of fertility compared to daily egg production. Fertility experiments typically focus on the ability of worms to produce viable offspring, while fecundity experiments are often used to assess simply egg hatching-related effects. We observed a decreasing trend in the total number of progeny upon exposure of 3D cuboid extracts (S1 Fig).

## *C. elegans* motility test: Effect of using 3D-printed cuboid-extracted water

The overall negative effects on lifespan, stress-response genes, and fertility were attributed to the toxic substances released into the media which contaminated *E. coli* bacteria. To investigate a different mode of toxicity effect, we employed a different extraction method by replacing LB media with distilled water (Fig 2C). This change in the extraction method provides a difference in the toxicity mode of action. The presence of toxic substances in water rather than in the food (i.e., *E. coli*) may induce the change of *C. elegans'* bending mobility. To test this, we selected approximately ten nematodes, all of which were age-synchronized to 3 days. They were then placed in either standard NGM or contaminated NGM by using the cuboid-extracted water. *C. elegans* is known to exhibit unique swimming behavior characterized by body bending in an aqueous environment. Therefore, any potential toxicity in the contaminated water might affects this behavior. After the exposure, the nematodes were removed from the NGM, rinsed with M9 buffer, and transferred to a 96-well plate. This setup allowed us to analyze the bending speeds of the two groups of *C. elegans*. We counted the number of body bends for 1 minute. Fig 2E shows time-lapse photo arrays representing body bending. For a healthy *C. elegans* worm, it takes approximately 1.6 sec to bend its body to an opposite position

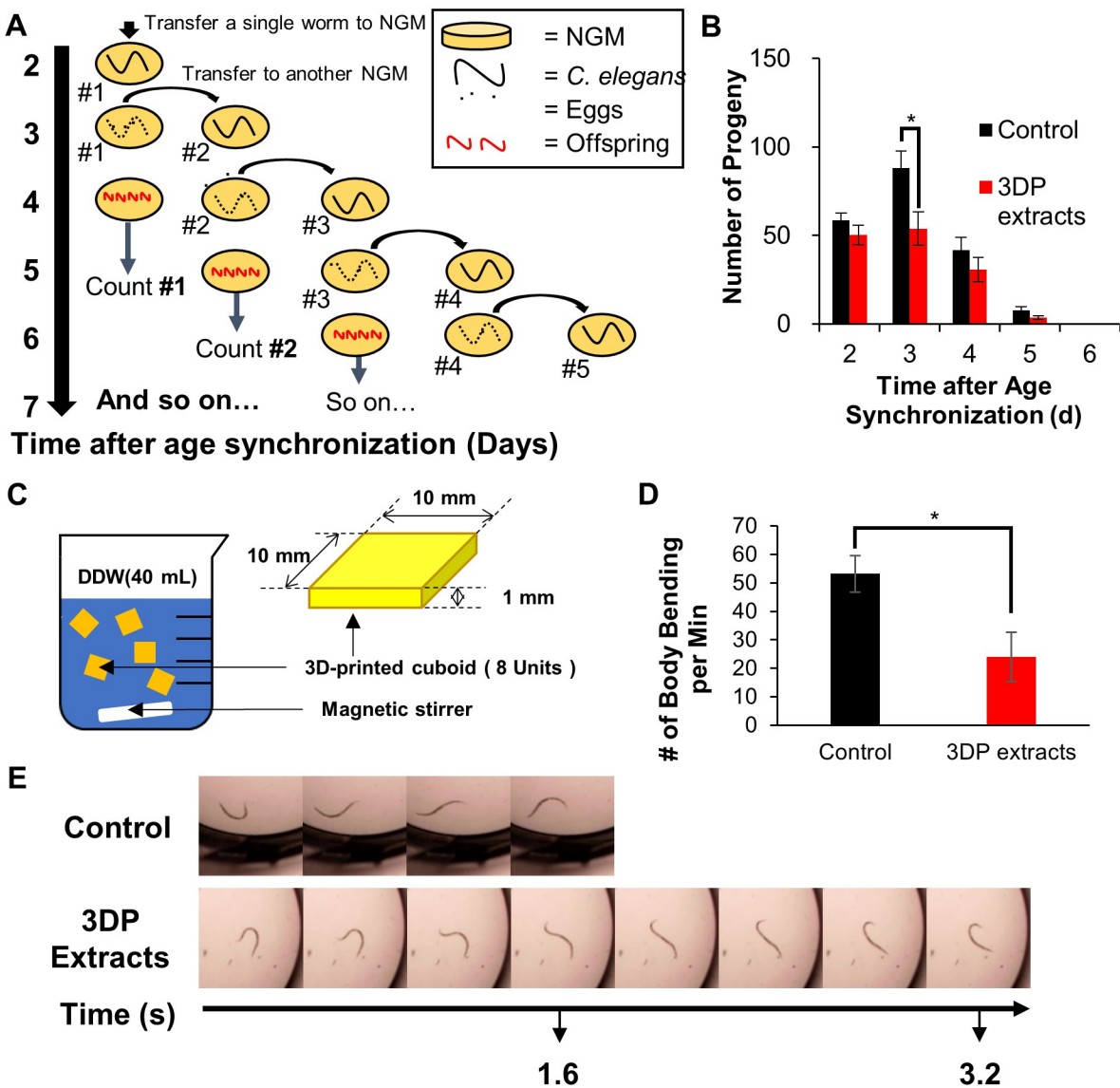

**Fig 2. Extraction procedure of 3D-printed cuboids in DDW and mobility decreases.** (A) Fertility assay procedure for *C. elegans*. (B) 3D-printed object extracts decrease fertility on day 3. Mean number of progeny produced by 5 individual worms each day. (C) Schematic description of extracting (72 hrs) toxic compounds from the 3D-printed cuboids in DDW (previously LB media). (D) The total number of observed body bending during 1 minute. The *C. elegans* exposed by 3DP extracts are red, and the ones by no exposure are black (n = 6). (E) Representative time-lapse photo arrays of body bending. The *C. elegans* exposed by 3DP extracts are shown in the lower raw, and the ones by no exposure are located in the upper raw. Asterisks indicate p-values less than 0.05 compared with the control, and the error bar indicates standard error.

(upper photos). However, we observed that twice the time was required for the same bending motion in the case of the exposed worms (lower photos). Quantitative analysis showed that the unexposed worms (control) bent 53.2 times per minute (n = 6), which was decreased to 24 bends per minute for the exposed *C. elegans* (n = 6) (Fig 2D). Although we cannot exclude the possibility of the OP50 taking up the toxins from the NGM media, these results show the adverse effect of contaminated water with the cuboid extracts on *C. elegans* motility.

### Identification of toxicity origin by X-ray photoelectron spectroscopy (XPS)

Biological toxicity of the photocurable resins comes from the photoinitiators [33]. However, the detailed information of the resin composition used in this study is quite limited due to the company's proprietary right. Nevertheless, we thought that the resin contains certain amount of photoinitiators. We hypothesized that unreacted photoinitiators would remain within the 3D-printed products until they leach out over time, potentially causing adverse effect on the health of *C. elegans*. Fig 3A illustrates the chemical structures of widely used UV-curable photoinitiators: 2-benzyl-2-(dimethylamino)-4'-morpholinobutyrophenone, 4-(dimethylamino)benzophenone, azobisisobutyronitrile, and 2-methyl-4-2-morpholinopropiophenone. These compounds have nitrogen atoms, which undergo homolytic cleavage upon exposure to light energy [34]. Consequently, the extracted solution contains the nitrogen-containing compounds. In fact, XPS analysis revealed that the nitrogen 1s signal was detected at 400.2 eV (Fig 3B), indicating that the observed toxicity of the extracted solution might originate from the nitrogen-containing photoinitiators. Further, we investigated the toxic effects of 2-methyl-4'-

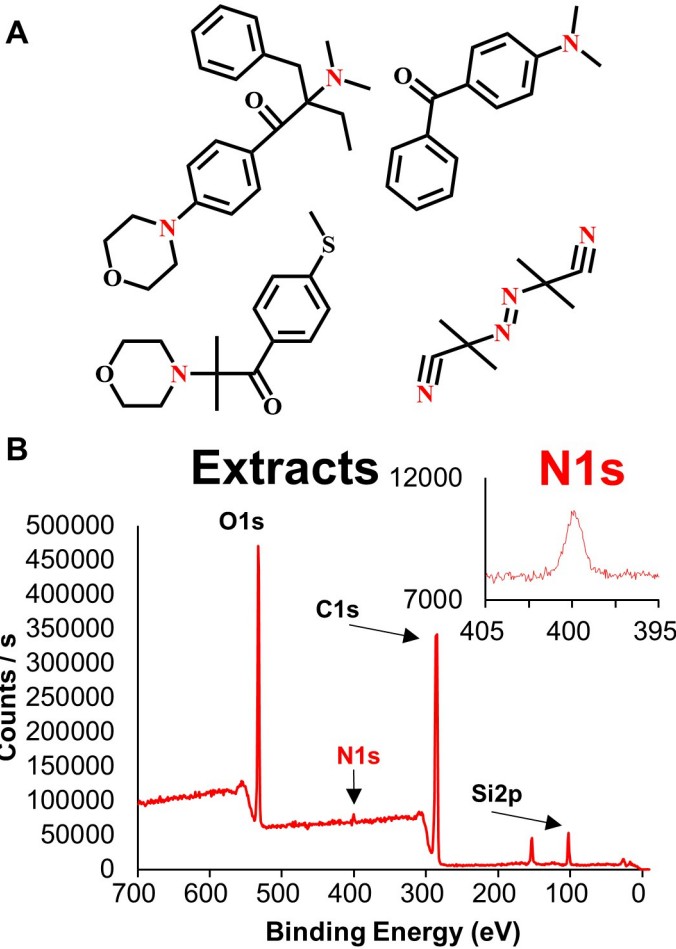

**Fig 3. Commonly used photoinitiators and XPS data of 3D-printed cuboid extracts.** (A) Chemical structures of commonly used photoinitiators. Clockwise from the top-left, 2-benzyl-2-(dimethylamino)-4'-morpholinobutyrophenone, 4-(dimethylamino)benzophenone, azobisisobutyronitrile, and 2-methyl-4-2-morpholinopropiophenone. Amine groups are marked in red. (B) XPS data of 3D-printed cuboid extracts. The inset plot is an enlargement of the N1s peak. XPS analysis showed the presence of nitrogen.

(methylthio)-2-morpholinopropiophenone, a widely used photoinitiator in 3D printing resins, on the developmental stage of *C. elegans*. The results showed that abnormal development began at the L3 stage, becoming more pronounced at the L4 stage, with smaller nematode sizes (red arrows) and unhealthy body shape (blue arrow) (S2 Fig). The concentration-dependent toxicity behavior showed that overall behavioral toxicity was barely observed at a 0.5 mM concentration, while toxicity onset occurred at 5 mM. This finding emphasizes the importance of considering local concentration of toxic compounds leaching out from a particular defect point of 3D printed solid materials when assessing toxicity as the effective concentration affecting toxicity in a homogeneously distributed solution may differ. This further highlights the utility of *C. elegans* as a unique model organism for detecting toxic compounds due to its combination of locomotion and sensing abilities.

## *C. elegans* is a unique animal model system with sensing and locomotion abilities

As mentioned in the introduction section, *C. elegans* possesses the ability to detect and move away from locations where harmful toxic compounds are present or being released, while maintaining its locomotion in non-toxic environments [15]. The chemotactic behavior displayed by *C. elegans* can be leveraged to identify the source of toxic compound release. In the context of the experiments described in Figs 1–3, which demonstrated that photoinitiators and other toxic compounds could be released from the 3D cuboids up to a certain toxicity level. Thus, we designed an experiment when fabricating a 3D cuboid. A cuboid with identical dimensions (1 x 1 x 0.1 cm$^3$) was printed, followed by UV light illumination for post-curing to minimize the release of toxic compounds. Simultaneously, during the post-curing step, we wrapped a piece of Al foil around one half of the cuboid (Fig 4A). Our hypothesis was that majority of toxic compounds would be released from the half block without post-UV curing. Indeed, the non-post-cured cuboid exhibited slight bending upon applying external force (Fig 4B).

We observed distinct behavioral patterns in individual *C. elegans* worms when they closely approached the non-post-cured side of the cuboid, rapidly turning away from the cuboid within less than 14 sec. (Fig 4D, upper photos). This 'turn-away' motion may be attributed to the formation of a gradient of toxic compounds released from the cuboid. We frequently observed this 'turn-away' action within a 3 mm distance from the edge of the cuboid. It is noteworthy that some *C. elegans* worms initially crossed over the 3 mm distance while approaching the cuboid, but then quickly turned away from it (lower photos). However, in contrast to the turning-away behavior observed on the non-post-cured side, *C. elegans* approaching the cured side (control) showed a different response. They moved closer to the cured side without displaying the rapid turning-away motion observed on the non-cured side. While occasional turns might still occur on the cured side, they were not as frequent as the ones observed on the uncured side.

Alongside these observations of the individual *C. elegans*, we also examined the overall population behavior. We monitored the geometrical distributions of all *C. elegans* organisms centered on the cuboid, one hour after worm introduction at the designated four points (Fig 4C). We opted for the 1 hr setting time to allow the release and gradient formation of toxic substances from the cuboids. We found that *C. elegans* worms were evenly distributed across both the Al foil masked or unmasked regions within a distance range from zero to 10 mm (Fig 4E, left bars, S1 Table). In other words, approximately 50% of the population of *C. elegans* was found in the masked region, with the remaining 50% located in the unmasked region. Upon further examination of the region from zero to 6 mm, however, we noticed differences in the

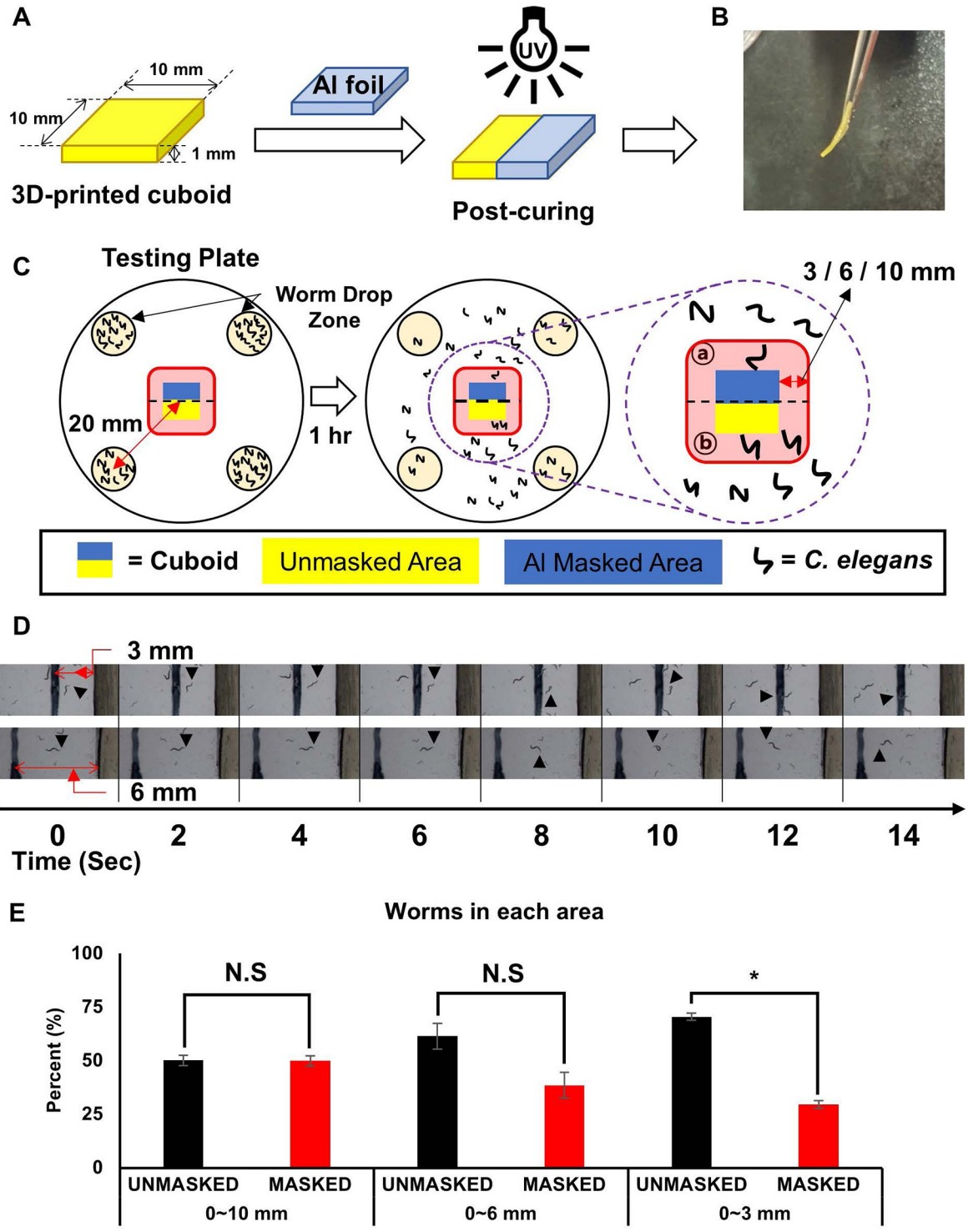

**Fig 4. Half area-cured 3D-printed material monitored in the presence of *C. elegans*.** (A) Schematic procedure for additional curing of the 3D-printed cuboid using aluminum foil masking (blue). (B) The picture of the additionally cured 3D-printed cuboid. (C) Experimental scheme of testing attraction or avoidance behavior of *C. elegans* toward either the additionally cured area or not. The cuboid experienced in the procedure (A) was placed at the center of the testing plate, and the worm was placed 20 mm aside from the center. (D) Representative time-lapse photo arrays of locomotion at 3 mm and 6 mm away from the 3D-printed cuboid. (E) Result of the additionally cured area detection assay using *C. elegans*. The proportion of worms in the unmasked area was higher than in the masked area, with statistical significance for the 0–3 mm area (p-value < 0.001). Asterisks indicate p-values less than 0.05 compared with the control, and the error bar shows standard error.

geometrical distributions. Approximately 61% of the *C. elegans* were found in the photocured region, with 39% located in the masked region (the middle bars) (p-value = 0.0531). This difference in chemotactic behavior became more pronounced when we observed the crawling worms within a range from zero to 3 mm from the cuboid. Approximately 70% of the *C. elegans* worms were located in the photocured region, and 30% in the masked region (the right bars) (p-value < 0.001). The results indicate that *C. elegans* can serve as a unique model organism for detecting toxic compounds because of its combination of locomotion and sensing abilities.

## Conclusions

Conventional model organisms have played a significant role in past toxicity assessments of various substances. However, these typical models lack the capability to accurately determine the direction of chemical diffusion from leachable materials. In this study, we employed 3D-printed products as representative leachable materials and utilized the small nematode *C. elegans*, which measures approximately 1 mm in body length, for our toxicity test. This approach enabled us to conduct a toxicity assessment that would have been challenging with traditional model organisms. Our findings showed that the extract obtained from the 3D-printed block had detrimental effects on the health of *C. elegans*, resulting in a shortened lifespan, reduced physical activity, decreased offspring count, and diminished expression levels of stress-response genes. Moreover, by tracking the movement of the worms in the vicinity of the substance, we were able to observe the leaching point and ascertain the direction of chemical diffusion of the toxic components. This research demonstrates the utility of *C. elegans* as a model organism, providing insights into the understanding and evaluation of material toxicity in a dynamic manner. Consequently, material toxicity assessments that were previously impractical with conventional model organisms can now be performed by monitoring the distribution and behavior of *C. elegans*.

## Supporting information

**S1 Fig. Total number of progeny.** The control group produced an average of 196 offspring, while *C. elegans* exposed to 3D-printed (3DP) extracts produced 138 offspring on average. Error bars represent standard error.
(PDF)

**S2 Fig. Effects of photoinitiators on early stage development of *C. elegans*.** "5 mM-M" represents 5 mM of 2-methyl-4′-(methylthio)-2-morpholinopropiophenone, and "0.5 mM-M" represents 0.5 mM of it. After L3 stage, "5 mM-M" showed abnormality of development. There were no significant differences observed between control and "0.5 mM-M" groups. Red bars indicate size of 1 mm.
(PDF)

**S1 Table. *C. elegans* distribution in each area.**
(PDF)

## Acknowledgments

All *C. elegans* strains were provided by Sang-Kyu Park (Department of Medical Biotechnology, College of Medical Science, Soonchunhyang University).

## Author Contributions

**Investigation:** Jun Sung Kim.

**Methodology:** Jun Sung Kim.

**Project administration:** Haeshin Lee.

**Resources:** Sang-Kyu Park.

**Supervision:** Haeshin Lee.

**Validation:** Haeshin Lee.

**Visualization:** Jun Sung Kim.

**Writing – original draft:** Jun Sung Kim.

**Writing – review & editing:** Haeshin Lee.

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
