## [Decision Letter · Decision Letter 0]

20 Mar 2023

PONE-D-23-04286Sniffer worm, C. elegans, as a toxicity evaluation model organism with sensing and locomotion abilitiesPLOS ONE

Dear Dr. Lee,

Thank you for submitting your manuscript to PLOS ONE. After careful consideration, we feel that it has merit but does not fully meet PLOS ONE’s publication criteria as it currently stands. Therefore, we invite you to submit a revised version of the manuscript that addresses the points raised during the review process.

 Please submit your revised manuscript by May 04 2023 11:59PM. If you will need more time than this to complete your revisions, please reply to this message or contact the journal office at plosone@plos.org. Please include the following items when submitting your revised manuscript:A rebuttal letter that responds to each point raised by the academic editor and reviewer(s). You should upload this letter as a separate file labeled 'Response to Reviewers'.A marked-up copy of your manuscript that highlights changes made to the original version. You should upload this as a separate file labeled 'Revised Manuscript with Track Changes'.An unmarked version of your revised paper without tracked changes. You should upload this as a separate file labeled 'Manuscript'.

We look forward to receiving your revised manuscript.

Kind regards,

Myon Hee Lee, Ph.D

Academic Editor

PLOS ONE

Journal Requirements:

   "This work was supported by a National Research Foundation of Korea (NRF) grant funded by the Korea government (MSIT) (No. 2018R1A5A1025208) and the Disaster and Safety Management Institute (MPSS-CG-2016-02) from The Ministry of Public Safety and Security of the Republic of Korea."

   "This work was supported by a National Research Foundation of Korea (NRF) grant funded by the Korea government (MSIT) (No. 2018R1A5A1025208) and the Disaster and Safety Management Institute (MPSS-CG-2016-02) from The Ministry of Public Safety and Security of the Republic of Korea (HL, JSK). The funders had no role in study design, data collection and analysis, decision to publish, or preparation of the manuscript."

Reviewers' comments:

Reviewer's Responses to Questions

**Comments to the Author**

1. Is the manuscript technically sound, and do the data support the conclusions?

Reviewer #1: Yes

Reviewer #2: Yes

2. Has the statistical analysis been performed appropriately and rigorously? 

Reviewer #1: No

Reviewer #2: Yes

3. Have the authors made all data underlying the findings in their manuscript fully available?

Reviewer #1: No

Reviewer #2: Yes

4. Is the manuscript presented in an intelligible fashion and written in standard English?

Reviewer #1: No

Reviewer #2: Yes

5. Review Comments to the Author

Reviewer #1: The study aims to investigate the possibility of using C. elegans as a model organism to assess toxicity of a material, in this case a 3D printed object. With increased use of 3D printing and the reported toxicity of the material, more study is needed to raise awareness and ensure people’s safety.

The authors used C. elegans lifespan, fecundity, expression of genes related to stress response, and motility, to assess the toxicity of a 3D printed product. In addition, using the aversive behavior C. elegans displays towards the toxins, the authors show that there is a localized avoidance behavior at one area of the product that produces more toxins, and quantifies this behavior.

The study makes a meaningful contribution to the field, especially at a time when new model organisms for toxicity tests are needed, to circumvent the issue of animal ethics in terrestrial vertebrates. In addition, although several model organisms have already been used to assess toxicity, C. elegans offers the unique potential to pinpoint the source of the toxins within the same material.

One glaring issue with the manuscript, however, is the writing. The manuscript will need a significant overhaul in the writing to provide clarity to the overall methods and results of the study.

Below are concerns that will need to be addressed before manuscript can be considered for publication:

1. Authors should seek the help of an English editing service to improve the quality of the writing.

2. There are also structural changes needed. For instance, detailed and lengthy description of methods in the Results section and the figure legends should move to the Methods section. The Results section should be reserved for discussing the results.

3. In the Abstract and Introduction section, the authors talk about “local defects” in 3D printed objects. However, for those not familiar with the 3D printing process, it is not clear why the defective area would release toxic compounds. Only upon reaching the X-ray photoelectron spectroscopy part of the Results section, it becomes clear that the defect probably refers to areas with unreacted photoinitiator compounds due to incomplete photocuring. I think this could be explained more clearly in the introduction for those not familiar with the resin 3D printing process.

4. 3D printed block extract is obtained in 2 ways: 1. Extraction with LB, which is then used to culture OP50, and 2. Extraction with water, which is added to the NGM media. Lifespan experiment uses #1. It is unclear which method is used for the GFP strains and fertility assay, because in the Methods section, it is stated that both experiments use “NGM plates supplemented with 3D-printed block extract” (line 135 and 144), which sounds like #2. However, in the text, both in the Methods section and the Results section, description of #2 comes after the two experiments are already mentioned. If those experiments are done using #1, the above quote should be switched to: “NGM plates seeded with contaminated OP50”.

5. The meaning of 3-day-old nematode, 5-day-old nematode is not clear (line 160, 162, 178). Does it refer to 3 day adult worms or 3 days from synchronized L1s? Please clarify, specifying the larval stage of the worm. If they are 5 day adult worms, they are rather aged, and the reason for using that particular age should be noted.

6. When describing the motility assay, at first glance the text seems to imply that the reason for extracting toxins in water is so that the worms can swim in them (line 293-295). But after careful reading, it seems that was a misunderstanding due to the unclear writing. However, even if the intention of the water extract was to add the toxins to the NGM surface rather than the food, this is also problematic, because OP50 continues to grow in the NGM media (NGM contains peptone) and could take up the toxins during the time from seeding to when worms are placed, plus the additional 2 days of culture. The only way to ensure the worms are only exposed to toxins through the agar surface is if OP50 is UV-killed or if NGM doesn’t contain peptone.

With that said, this is a minor distinction and does not affect the overall conclusion of the study. Therefore, it should suffice to add a disclaimer sentence, such as: “..although we cannot exclude the possibility of the OP50 taking up the toxins from the NGM media..”

7. Also, it is not clear why the motility test should be done with a different exposure method. Please explain.

8. When explaining the aversive behavior (Figure 4D), the authors should describe the behavior at the cured control side to contrast with the “turning-away” behavior observed in the uncured side. Are these observed in both sides, but just in differing degrees? Or are there no turning away behavior observed in the cured side?

9. For Fig 4E, authors should also provide the raw numbers used to calculate the percentage, so that readers may know how many worms are used to assess the trend. Also, method of statistical analysis should be mentioned.

10. The blue and yellow arrows in Fig 4E are not necessary.

Reviewer #2: In this manuscript, Kim et al. present C. elegans, an animal model system, for toxicity evaluation using sensing and locomotion abilities for three-dimensional (3D) printing(3DP) extracts. The authors used C. elegans as an animal model to study the effects of 3DP extracts and observed that these extracts led to a reduction in lifespan, stress-gene expression, reproduction, and motility of the nematodes. The manuscript shows potential interest; however, some points need clarification:

Major Concerns:

1. The author needs to specify the choice of the used FUdR, as it is known to affect the differential expression of many genes.

2. In online 143, the author mentions that "five L4/young adult stage worms were transferred to a fresh NGM plate containing 3D-printed block extract and permitted to lay eggs for 5hr." However, L4 stage worms are known to produce eggs after at least 8 hours. The authors should explain the egg-laying time.

3. The authors measured the number of eggs after the treatment of 3D-printed object extracts. Instead of showing produced daily eggs, the authors should show total progeny.

4. The authors should demonstrate the treatment's effect on the development stage and adult lethality for toxic effects in C. elegans."

6. PLOS authors have the option to publish the peer review history of their article (what does this mean?). If published, this will include your full peer review and any attached files.

Reviewer #1: No

Reviewer #2: No

---

## [Author Response · Author response to Decision Letter 0]

30 Apr 2023

Reviewer #1: The study aims to investigate the possibility of using C. elegans as a model organism to assess toxicity of a material, in this case a 3D printed object. With increased use of 3D printing and the reported toxicity of the material, more study is needed to raise awareness and ensure people’s safety.

The authors used C. elegans lifespan, fecundity, expression of genes related to stress response, and motility, to assess the toxicity of a 3D printed product. In addition, using the aversive behavior C. elegans displays towards the toxins, the authors show that there is a localized avoidance behavior at one area of the product that produces more toxins, and quantifies this behavior.

The study makes a meaningful contribution to the field, especially at a time when new model organisms for toxicity tests are needed, to circumvent the issue of animal ethics in terrestrial vertebrates. In addition, although several model organisms have already been used to assess toxicity, C. elegans offers the unique potential to pinpoint the source of the toxins within the same material.

One glaring issue with the manuscript, however, is the writing. The manuscript will need a significant overhaul in the writing to provide clarity to the overall methods and results of the study.

Below are concerns that will need to be addressed before manuscript can be considered for publication:

1. Authors should seek the help of an English editing service to improve the quality of the writing.

<Answer>

Thank you for your helpful advice. We acknowledged that there are some sentences that can be misunderstood by readers because of the quality of the writing. We have revised the abstract to enhance clarity, precision, and scientific rigor, while also refining the language for conciseness. Additionally, we have improved the introduction by making it more concise and using clear language. The revised manuscripts are the followings.

- Original abstract [Page 1, line 17 - Page 2, line 31]

The probability of objects fabricated by three-dimensional (3D) printing exhibiting local defects is higher than that detected in products of conventional casting-based manufacturing. Multistep layer-by-layer procedures in additive manufacturing are the main reason. Light intensity and/or penetration depth, inhomogeneity of components, and variations in nozzle temperature are factors that create local defects. Defect regions are sources of toxic component release, but methods to identify them in printed materials have not been reported. Existing assays for evaluating material toxicity are based on extraction, and these toxicological assays use living creatures to passively detect harmful agents in extracted solutions. Thus, the development of an active system for identifying sites of toxicity sources is a critical and urgent issue in 3D printing technologies. Herein, we introduce an animal model system, C. elegans, for toxicity evaluation. C. elegans crawls toward safe regions but avoids toxically dangerous areas. The ‘sensing’ and ‘locomotion’ abilities of C. elegans are unparalleled among existing underwater and animal models, providing immediate indications to help find toxicity source sites.

- Revised abstract [Page 1, line 17 - Page 2, line 31]

Three-dimensional (3D) printed objects have a higher likelihood of containing local defects compared to products manufactured by conventional casting-based methods. The multistep, layer-by-layer process in additive manufacturing, with factors such as light intensity, variations of light penetration depth, inhomogeneity of components, and fluctuation of nozzle temperature, contribute to the increased defect generations. Defect regions are sources of toxic component release, but methods to identify them in printed materials have not been reported. Traditional toxicological assays rely on extraction followed by treatments to living creatures, which is a passive detection of harmful agents. Thus, the development of an active system to identify and locate the sources of toxicity is a critical issue in 3D printing technologies. Herein, we present the use of the nematode model system, C. elegans, for toxicity evaluation. C. elegans displays unique ‘sensing’ and ‘locomotion’ abilities, actively moving toward safe regions while avoiding toxic areas. These capabilities make C. elegans an unparalleled choice among existing aquatic and animal models, offering immediate and precise indications to locate and identify the sources of toxicity in 3D printed materials.

- Original introduction [Page 2, line 34 - Page 4, line 96]

One critical and unavoidable step in developing new materials for biomedical applications is the toxicological evaluation of newly developed materials [1]. Living organisms must be chosen when evaluating material toxicity. Early studies in toxicological tests utilized plant species such as Lemna gibba [2]. This plant is small in size, easily grown, and short in life cycle [3, 4]. An advantage of this approach is that growth inhibition by any toxic components can be easily observed. However, the results of Lemna gibba growth inhibition are not directly related to toxicological effects on humans.

Due to the aforementioned limitations related to correlation with human toxicity levels, later studies have introduced animal models. Two categories have been developed. One involves underwater organisms, such as Daphnia [5, 6] or Danio rerio (zebrafish) [7, 8] and the other is based on terrestrial animals, for instance, a rat or a mouse [9]. Toxicological outcomes observed in underwater animals are expressed by one or more factors via the following: acute immobilization, mobility, survival, developmental disorder, and/or embryo hatching [6]. However, the results obtained in underwater environments are very different from those obtained in ambient conditions. Thus, terrestrial animal models have become popular. Rats and mice have been dominantly chosen as animal models recently because they are small and inexpensive compared to other mammals and produce many descendants. However, strict and wide-ranging ethical regulations have recently become a major obstacle to the use of model systems [10]. For this reason, organ-on-a-chip or related technologies have been introduced [11], and cosmetic science has just begun to find an alternative method to test the toxicity of newly introduced components.

The aforementioned challenges in current model systems motivate researchers to develop a new convenient and effective way to address this ethical issue. Importantly, additive manufacturing technologies known as 3D printing also require a novel toxicity evaluation system [12]. Compared to those manufactured by a conventional casting-based method, objects prepared by 3D printers intrinsically present many defective regions. When building up an object, additive layer-by-layer deposition processes essentially go through many repeated steps during which unavoidable defects are generated. For stereolithography (SLA) methods, light intensity and/or penetration depth, inhomogeneity in polymers concentrations, and dispersion of components can slightly vary [13]. For fused deposition modeling (FDM), variations in nozzle temperature in systems are a factor that create local defects when printing an object [14]. As 3D printing techniques are suitable for small-quantity prototype production, each product exhibits varied sites of defects that result in a different level of toxicity.

Therefore, the development of a new animal model that detects product-to-product variations in toxicity sources is necessary. Unfortunately, the aforementioned plant and animal models cannot satisfy this need (Table 1). In the case of plant models, the living creature has no locomotive function to identify sites of toxicological sources in printed materials. The underwater toxicity evaluation system, zebrafish, has a locomotive function, but leached toxic compounds are diffused in water. Thus, zebrafish are not a suitable model system for identifying the toxicity sources in printed materials. Foreign body reactions to implanted materials in mice and rats also make it difficult to identify toxicity sources.

For the aforementioned reasons, the combination of sensing and locomotion is an important factor in the development of a new toxicity model. Here, we propose Caenorhabditis elegans (C. elegans) as a model that can detect both overall toxicity and toxic sites of printed materials. The reason for choosing C. elegans is that it essentially exhibits chemotactic behaviour [15]. The crawling behavior of C. elegans provides phenotypic indications for detecting sources of toxicity origins in a test material. In addition, the lifespan of C. elegans is from two to four weeks, which is suitable for a model system [16]. C. elegans is one of a few creatures where all its genome sequences are known [17]. The body length of the adult worm is approximately 1 mm, and all differentiation pathways of 959 cells in total from a single germinal cell are reported, which also makes C. elegans suitable for evaluating toxicity effects [18]. This attractive animal has already been studied in various fields, such as aging, new drug development, and neurobiology [19-23]. However, considerations and attempts to utilize C. elegans as an ‘active’ toxicological model with sensing and locomotion abilities have never been studied.

- Revised introduction [Page 2, line 34 - Page 4, line 97]

A critical and indispensable step in developing new materials for biomedical applications is the toxicological evaluation of the novel materials [1]. The assessment necessitates the use of living organisms to determine material toxicity. Early toxicological studies employed plant species, such as Lemna gibba [2]. This plant’s small size, easy of cultivation, and short life cycle make it a practical choice for toxicity evaluations [3, 4]. Primary advantage of using this plant lies in the ease of observing growth inhibition by toxic components. However, the results obtained from Lemna gibba growth inhibition did not directly correspond to toxicological effects on humans.

Due to the limitations associated with the correlation between plant-based toxicity models and human toxicity levels, later studies have introduced animal models. These models can be broadly categorized into two groups. One involves aquatic organisms, such as Daphnia [5, 6] or Danio rerio (zebrafish) [7, 8] and the other is terrestrial animals, including rats or mice [9]. Toxicological effects observed in aquatic animals are assessed by various factors, such as acute immobilization, mobility, survival, developmental disorder, and/or embryo hatching [6]. However, the results obtained in aquatic environments are very different from those in ambient conditions, leading to a preference for terrestrial animal models. Rats and mice have been dominantly chosen as animal models due to small size, affordability, and high reproductive rates. However, implementation of stringent and wide-ranging ethical regulations has recently posed a significant challenge to the use of these model systems [10]. Consequently, organ-on-a-chip technologies have been introduced as alternative approaches [11]. Additionally, cosmetic science has started to find new methods for assessing the toxicity of novel components.

The challenges associated with current model systems have motivated researchers to develop convenient and ethically sound methods for toxicity evaluation. Importantly, additive manufacturing technologies known as 3D printing also necessitate the development of new toxicity assessment systems [12]. Compared with objects manufactured through a conventional casting-based method, ones produced by 3D printing inherently exhibit increased defective regions. Reasons include that layer-by-layer deposition processes in additive manufacturing essentially involves numerous repeated steps, during which unavoidable defects are generated. In stereolithography (SLA) methods, variations in light intensity, penetration depth, and inhomogeneity of polymers concentrations and dispersion can occur [13]. In fused deposition modeling (FDM), nozzle temperature fluctuations contribute to the formation of local defects during printing [14]. As 3D printing techniques are ideal for small-scale prototype production, each product may display unique sites of defects, resulting in varying levels of toxicity.

Thus, there is a need to develop a new animal model capable of detecting product-to-product variations in toxicity sources. Unfortunately, the existing plant and animal models are insufficient for this purpose (Table 1). Plant models lack the locomotive function to identify toxicological sources within printed materials. While zebrafish, an aquatic toxicity evaluation model, has a locomotive function, leached toxic compounds become diffused in water, making them unsuitable for pinpointing toxicity sources in printed materials. Additionally, foreign body reactions to implanted materials in mice and rats further complicate identification of toxicity sources.

In light of these challenges, a new toxicity model should combine both sensing and locomotion capabilities. Here, we propose the use of Caenorhabditis elegans (C. elegans) as a model capable of detecting both the overall toxicity and specific toxic sites within printed materials. C. elegans was chosen due to its inherent chemotactic behaviour [15]. The nematode’s crawling activity serves as phenotypic indicators for identifying the sources of toxicity within test materials. Furthermore, the lifespan of C. elegans is short from two to four weeks, making it a suitable model system [16]. As one of the few organisms with a fully sequenced genome [17], the adult worm measures approximately 1 mm in length, with well-documented differentiation pathways for all 959 cells originating from a single germinal cell [18]. This feature makes C. elegans an ideal candidate for evaluating toxicity effects. This attractive animal has already been studied in various fields, including aging, new drug development, and neurobiology [19-23]. However, the potential of C. elegans as an ‘active’ toxicological model, incorporating both sensing and locomotion abilities, has not been explored previously.

2. There are also structural changes needed. For instance, detailed and lengthy description of methods in the Results section and the figure legends should move to the Methods section. The Results section should be reserved for discussing the results.

<Answer>

Thank you very much for your helpful advice. The reviewer’s comment is right because we found lengthy statement particularly in the figure legends. The revised and shortened the legends are the followings.

- Original figure legend [Fig 1.]

Fig 1. Procedure of 3D-printed cuboids in LB and the effect of extracts on C. elegans health. (A) As shown in the scheme, the cuboid was printed in dimensions of 1 x 1 x 0.1 cm3. A total of 80 cuboids and a magnetic stirrer were added to 400 mL of LB media. After the bottle was blocked from light and extracted at room temperature for 72 hours, the following experiments were performed under the corresponding conditions. (B) Lifespan-reducing effect of 3D-printed cuboid extracts. Dotted lines indicate 50% survival of C. elegans. Age-synchronized young adult worms (day 3, n=60) were transferred to NGM plates layered with OP50, which was grown in LB extract, and the number of live/dead worms was recorded every day. Worms that were lost or showed internal hatching during the assay were excluded. The mean lifespan was observed and compared with that of worms grown with OP50 under normal non-toxic conditions, control (p<0.05). (C) The stress-response gene expression levels: HSP-16.2::GFP (upper panels) SOD-3::GFP (lower panels). The left column indicates uncontaminated LB supplementation, and the right column is the LB contaminated with released toxic compounds. (D) Quantitative analysis of the stress-responsive protein expression levels shown in Fig 1C. The black bars are the results obtained using uncontaminated LB supplementation, and the red bars are the results obtained using the LB contaminated with released toxic compounds. The asterisks indicate p-values less than 0.05 compared with the control.

- Revised figure legend [Fig 1.]

Fig 1. Procedure of 3D-printed cuboids in LB and the effect of extracts on C. elegans health. (A) Schematic description of extracting toxic compounds from the 3D-printed cuboids. (B) Lifespan reduction of C. elegans by 3D-printed cuboid extracts (red line). Dotted lines indicate 50% survival of C. elegans. (C) Stress-response gene expression levels: HSP-16.2::GFP (upper panels) SOD-3::GFP (lower panels) in response to LB extract with or without released toxic compounds. (D) Quantitative analysis of the stress-responsive protein expression levels shown in Fig 1C. The black bars are the results obtained using uncontaminated LB supplementation, and the red bars are the results obtained using the LB contaminated with released toxic compounds. The asterisks indicate p-values less than 0.05 compared with the control.

- Original figure legend [Fig 2.]

Fig 2. Extraction procedure of 3D-printed cuboids in DDW and mobility decreases. (A) Schematic procedure for the fertility assay of C. elegans. (B) 3D-printed object extracts decrease fertility on day 3. The number of progeny produced was monitored every day during a gravid period. Data show the mean number of progeny produced by 5 individual worms on each day. Asterisks indicate p-values less than 0.05 compared with the control. (C) Extraction procedure of 3D-printed blocks in DDW (previously LB media). After a 72 hr extraction step, the extract was filtered through a 0.2 µm pore syringe filter. (D) Total number of body bends in 1 minute. The control (black) bent 53.16 times per minute, while worms exposed to 3D-printed cuboid extract (red) bent 24 times per minute (n=6). Asterisks indicate p-values less than 0.05 compared with the control, and the error bar indicates standard error. (E) Representative time-lapse photo arrays of body bending. Compared to the control worm, the worm treated with the 3D-printed cuboid extract took twice as much time to turn in the opposite direction.

- Revised figure legend [Fig 2.]

Fig 2. Extraction procedure of 3D-printed cuboids in DDW and mobility decreases. (A) Fertility assay procedure for C. elegans. (B) 3D-printed object extracts decrease fertility on day 3. Mean number of progeny produced by 5 individual worms each day. (C) Schematic description of extracting (72 hrs) toxic compounds from the 3D-printed cuboids in DDW (previously LB media). (D) The total number of observed body bending during 1 minute. The C. elegans exposed by 3DP extracts are red, and the ones by no exposure are black (n=6). (E) Representative time-lapse photo arrays of body bending. The C. elegans exposed by 3DP extracts are shown in the lower raw, and the ones by no exposure are located in the upper raw. Asterisks indicate p-values less than 0.05 compared with the control, and the error bar indicates standard error.

- Original figure legend [Fig 4.]

Fig 4. Half area-cured 3D-printed material monitored in the presence of C. elegans. (A) Scheme of additional curing of a 3D-printed cuboid. Half of the cuboid is wrapped with aluminum foil (used as photomask, blue), and then additional curing is performed using a UV curing machine for 1 minute. (B) Picture of the result of additional UV curing of a 3D-printed cuboid. The area that was masked by Al foil was relatively softer than the area that was not. (C) Design of the testing plate for an additionally photocured area detection assay using C. elegans. The apricot color is the ‘worm drop zone’, which indicates where C. elegans worms are placed at the beginning of the experiment. We tested whether the worms could distinguish between two areas, the additionally masked area and the unmasked area. The object resulting from the procedure in (A) was placed at the center of the testing plate, and the worm drop was placed 20 mm from the center. After placing the worm drop, the worms were incubated at room temperature for 1 hr. Then, the number of worms in the designated area was counted. (D) Representative time-lapse photo arrays of locomotion. The top shows the area 3 mm away from the 3D-printed cuboid, and the bottom is the area 6 mm away from the cuboid. (E) Result of the additionally photocured area detection assay using C. elegans. Regarding the middle bars, which include worms in the area within 0 to 6 mm, 61.42% of the worms were in the unmasked area, and 38.57% of the population was in the masked area (p-value = 0.0531). Regarding the right bars, which include the worms in the 0-3 mm area, there were 70.45% of the worms in the unmasked area and 29.54% in the masked area (p-value < 0.001). Colored arrows (blue, yellow) are marked to show trends. Asterisks indicate p-values less than 0.05 compared with the control, and the error bar indicates standard error.

- Revised figure legend [Fig 4.]

Fig 4. Half area-cured 3D-printed material monitored in the presence of C. elegans. (A) Schematic procedure for additional curing of the 3D-printed cuboid using aluminum foil masking (blue). (B) The picture of the additionally cured 3D-printed cuboid. (C) Experimental scheme of testing attraction or avoidance behavior of C. elegans toward either the additionally cured area or not. The cuboid experienced in the procedure (A) was placed at the center of the testing plate, and the worm was placed 20 mm aside from the center. (D) Representative time-lapse photo arrays of locomotion at 3 mm and 6 mm away from the 3D-printed cuboid. (E) Result of the additionally cured area detection assay using C. elegans. The proportion of worms in the unmasked area was higher than in the masked area, with statistical significance for the 0-3 mm area (p-value < 0.001). Asterisks indicate p-values less than 0.05 compared with the control, and the error bar shows standard error.

3. In the Abstract and Introduction section, the authors talk about “local defects” in 3D printed objects. However, for those not familiar with the 3D printing process, it is not clear why the defective area would release toxic compounds. Only upon reaching the X-ray photoelectron spectroscopy part of the Results section, it becomes clear that the defect probably refers to areas with unreacted photoinitiator compounds due to incomplete photocuring. I think this could be explained more clearly in the introduction for those not familiar with the resin 3D printing process.

<Answer>

Thank you for your insightful advisory. As you recommended, the motivation part written in the X-ray photoelectron spectroscopy in the result section was briefly added. First, we revised the corresponding paragraph of the part explaining that ‘the repeated numerous steps inherently increase defect sites’ for better understanding.

- Original manuscript [Page 3, line 57 - 69]

The aforementioned challenges in current model systems motivate researchers to develop a new convenient and effective way to address this ethical issue. Importantly, additive manufacturing technologies known as 3D printing also require a novel toxicity evaluation system [12]. Compared to those manufactured by a conventional casting-based method, objects prepared by 3D printers intrinsically present many defective regions. When building up an object, Additive layer-by-layer deposition processes essentially go through many repeated steps during which unavoidable defects are generated. For stereolithography (SLA) methods, light intensity and/or penetration depth, inhomogeneity in polymers concentrations, and dispersion of components can slightly vary [13]. For fused deposition modeling (FDM), variations in nozzle temperature in systems are a factor that create local defects when printing an object [14]. As 3D printing techniques are suitable for small-quantity prototype production, each product exhibits varied sites of defects that result in a different level of toxicity.

- Revised manuscript [Page 3, line 57 - 72]

The challenges associated with current model systems have motivated researchers to develop convenient and ethically sound methods for toxicity evaluation. Importantly, additive manufacturing technologies known as 3D printing also necessitate the development of new toxicity assessment systems [12]. Compared with objects manufactured through a conventional casting-based method, ones produced by 3D printing inherently exhibit increased defective regions. Reasons include that layer-by-layer deposition processes in additive manufacturing essentially involves numerous repeated steps, during which unavoidable defects are generated. In stereolithography (SLA) methods, variations in light intensity, penetration depth, and inhomogeneity of polymers concentrations and dispersion can occur [13]. In fused deposition modeling (FDM), nozzle temperature fluctuations contribute to the formation of local defects during printing [14]. As 3D printing techniques are ideal for small-scale prototype production, each product may display unique sites of defects, resulting in varying levels of toxicity.

In addition, we added a description of toxicity of photoinitiators which are indispensable components in SLA 3D printing used in this study. Toxicity occurs when photoinitiators are split into two pieces exhibiting radical species. 

- Revised manuscript [Page 3, line 66 - 68]

~dispersion can occur. Furthermore, the irradiation with UV-light leads to a homolytic bondage cleavage and generation of two highly reactive radical species, which are the origin of chemical toxicity. In fused deposition modeling~

4. 3D printed block extract is obtained in 2 ways: 1. Extraction with LB, which is then used to culture OP50, and 2. Extraction with water, which is added to the NGM media. Lifespan experiment uses #1. It is unclear which method is used for the GFP strains and fertility assay, because in the Methods section, it is stated that both experiments use “NGM plates supplemented with 3D-printed block extract” (line 135 and 144), which sounds like #2. However, in the text, both in the Methods section and the Results section, description of #2 comes after the two experiments are already mentioned. If those experiments are done using #1, the above quote should be switched to: “NGM plates seeded with contaminated OP50”.

<Answer>

Thank you for pointing out that the author wrote a sentence that could be misleading. The GFP strains and fertility assay were performed with the same method to lifespan experiment. We have made the following changes based on your advice.

- Original manuscript [Page 6, line 134 - 135]

Age-synchronized CL2070 (hsp-16.2::gfp) and CF1553 (sod-3::gfp) worms were placed in NGM plates supplemented with 3D-printed block extract at 20 °C for 5 days.

- Revised manuscript [Page 6, line 135 - 136]

Age-synchronized CL2070 (hsp-16.2::gfp) and CF1553 (sod-3::gfp) worms were placed in NGM plates seeded with contaminated OP50 at 20 °C for 5 days.

- Original manuscript [Page 6, line 143 - Page 7, line 146]

For the fertility assay, five L4/young adult stage worms were transferred to a fresh NGM plate containing 3D-printed block extract and permitted to lay eggs for 5 hrs. The eggs were maintained at 20 °C for 2 days. A single worm was transferred to a fresh NGM plate containing 3D-printed block extract every day until it laid no eggs.

- Revised manuscript [Page 6, line 143 - 146]

For the fertility assay, five late L4 or early young adult stage worms were transferred to a fresh NGM plates seeded with contaminated OP50 and permitted to lay eggs for 5 hrs. The eggs were maintained at 20 °C for 2 days. A single worm was transferred to a fresh NGM plates seeded with contaminated OP50 every day until it laid no eggs.

5. The meaning of 3-day-old nematode, 5-day-old nematode is not clear (line 160, 162, 178). Does it refer to 3 day adult worms or 3 days from synchronized L1s? Please clarify, specifying the larval stage of the worm. If they are 5 day adult worms, they are rather aged, and the reason for using that particular age should be noted.

<Answer>

Thank you for your guidance. The meaning of 3-day-old nematode indeed indicates 3 day young adult after egg laying, not 3 days from synchronized L1s. As reported by Byerly et al. (1976) at 20 °C, C. elegans undergoes the fourth molt approximately 56 hours after egg laying and enters the young adult stage [1]. In our study, we conducted experiments on the third day after egg laying, which corresponds to mostly late L4 or young adult worms. We did not used 5 day adult worm. Rather, 2 additional days of toxin exposure resulted in 5-day-old nematodes. We have revised the sentences to improve better understanding and to reduce potential confusion of the readers.

- Original manuscript [Page 7, line 159 - 162]

After that, we transferred a 3-day-old nematode to the prepared NGM. Before the start of this assay, a worm was put on a sterile fresh NGM agar plate without OP50 and allowed to crawl freely to remove the agglomerated bacteria from the worms (5 days old).

- Revised manuscript [Page 7, line 159 - 164]

After that, we transferred a 3-day-old nematode (i.e., the 3rd day from egg laying, corresponding to L4/young adult stage) to the prepared NGM. Before the start of this assay, 5-day-old worms (i.e., the 5th day from egg laying, corresponding to adult stage) were put on a sterile fresh NGM agar plate without OP50 and allowed to crawl freely to remove the agglomerated bacteria from them.

- Original manuscript [Page 8, line 177 - 179]

After that, we put the produced block in the center of the testing plate and put C. elegans worms at the same age (3 days old) in four places 20 mm away from the center (Fig 4C).

- Revised manuscript [Page 8, line 179 - 182]

After that, we put the produced block in the center of the testing plate and put C. elegans worms at the same age of 3-day-old (i.e., the 3rd day from egg laying, corresponding to L4/young adult stage) in four places 20 mm away from the center (Fig 4C).

[The Authors’ response continues]

Firstly, exposing worms to toxins at a very young age (e.g., L1 or L2) can interfere with their normal development and potentially unexpected error results. By waiting until the worms have reached a later developmental stage, we can ensure that they have developed normally before subjecting them to toxins. Secondly, adult stage worms were used in this experiment as they are more feasible and easier to handle than younger stages such as L1-L4. In Addition, observation of their body bending is difficult if the worms are too young to small in size (i.e., L1).

6. When describing the motility assay, at first glance the text seems to imply that the reason for extracting toxins in water is so that the worms can swim in them (line 293-295). But after careful reading, it seems that was a misunderstanding due to the unclear writing. However, even if the intention of the water extract was to add the toxins to the NGM surface rather than the food, this is also problematic, because OP50 continues to grow in the NGM media (NGM contains peptone) and could take up the toxins during the time from seeding to when worms are placed, plus the additional 2 days of culture. The only way to ensure the worms are only exposed to toxins through the agar surface is if OP50 is UV-killed or if NGM doesn’t contain peptone.

With that said, this is a minor distinction and does not affect the overall conclusion of the study. Therefore, it should suffice to add a disclaimer sentence, such as: “..although we cannot exclude the possibility of the OP50 taking up the toxins from the NGM media..”

<Answer>

Thank you for your comment. We acknowledged that the OP50 bacteria, which continue to grow in the NGM media containing peptone, could potentially take up the toxins during the time from seeding to when worms are placed, as well as during the additional 2 days of culture. So, we revised the manuscript as following.

- Original manuscript [Page 13, line 304 - 306]

Thus, the overall motility test clearly demonstrated the adverse motility effect of the use of contaminated water produced from cuboid extraction.

- Revised manuscript [Page 13, line 300 - 303]

Although we cannot exclude the possibility of the OP50 taking up the toxins from the NGM media, these results clearly demonstrate the adverse effect of contaminated water produced from cuboid extraction on C. elegans motility.

7. Also, it is not clear why the motility test should be done with a different exposure method. Please explain.

<Answer>

Thank you for your comment. In our previous experiments using LB media, we observed negative effects on lifespan, stress-response genes, and fertility in C. elegans. These results were likely due to the presence of toxic substances within the contaminated OP50, which was prepared in the LB media. To specifically investigate the impact of the water itself, we employed a different extraction method by replacing the LB media with distilled water. 

We hypothesized that the presence of toxic substances in the water might affect the bending mobility of C. elegans, which swims by bending its body in an aqueous solution. To test this hypothesis, we exposed approximately ten age-synchronized C. elegans nematodes to either regular NGM or contaminated NGM generated from cuboid-extracted water. After exposure, the worms on NGM were washed with M9 buffer and transferred to a 96-well plate to observe the bending speed of each group of C. elegans. This approach allowed us to better understand the direct effects of toxic substances in the extracted water on C. elegans motility. 

8. When explaining the aversive behavior (Figure 4D), the authors should describe the behavior at the cured control side to contrast with the “turning-away” behavior observed in the uncured side. Are these observed in both sides, but just in differing degrees? Or are there no turning away behavior observed in the cured side?

<Answer>

Thank you for your comment. The authors indeed observed the behavior at the both cured and uncured sides. However, before addressing your question about “turning-away” behavior, we would like to explain the difference of total population between the cured and uncured sides. The reason for mentioning the total population difference is that the side of releasing toxic components provides harsh condition for living. In other words, decreased number of the worm should be observed at the side of releasing toxic components. This answer as closely related to the next question #9.

 Distance from surface of the object 1st 2nd 3rd Average

Unmasked

(cured) 3mm 11 12 29 17.3

 6mm 18 19 63 33.3

 10mm 45 38 59 47.3

 Total 74 69 151 98

Masked

(uncured) 3mm 4 5 14 7.7

 6mm 10 19 27 18.7

 10mm 40 46 54 46.7

 Total 54 70 95 73

S1 Table. C. elegans distribution in each area.

S1 Table presents the results of our observation of the number of C. elegans at varying distances from the cuboid surface. Therefore, the number of worms on the cured and uncured sides differed depending on the distance from the edge of cuboid. Within 3 mm from edge of the cured side of the cuboid, an average of 17.3 worms were observed, while an average of 7.7 worms were present on the uncured side. At 6 mm, 33.3 worms were observed on the cured side, which was decreased down to 18.7 on the uncured side. At 10 mm, we observed 47.3 and 46.7 worms on the cured and uncured sides, respectively. As the distance from the edge increased beyond 10 mm, the difference in worm number found on both of the cured and uncured sides became a similar level because of dilution of toxic chemicals. Thus, both the number of worms and their “turning-away” behavior are important indicators.

As for the “turning-away” issue on cured side, our observation is the following. In contrast to the turning-away behavior observed in the uncured side, C. elegans on the cured side displayed a different response. The worms approached and moved closer to the cured side without displaying the rapid turning-away motion seen in the uncured side. While occasional turns might still occur on the cured side, they were not as frequent or distinct as the ones observed on the uncured side. This difference in behavior suggests that the worms sensed lower levels of toxic compounds in the cured sides, allowing them to move closer without avoiding the area. Therefore, the aversive behavior was much more frequently observed in the uncured side, while the cured side demonstrated less avoidance, indicating a significant difference in the perception of toxicity between the two sides. 

We revised manuscript to provide a clearer understanding of the aversive behavior observed in Fig 4D.

- Original manuscript [Page 15, line 348 - 354]

We observed a particular C. elegans worm that closely approached the side of the cuboid without post-curing and then rapidly (< 14 sec) turned away from the cuboid. This ‘turn-away’ motion might be due to formation of a gradient of toxic compounds released from the cuboids (Fig 4D, upper photos). This ‘turn-away’ motion was often found within a 3 mm distance from the edge side of the cuboid. Another case we observed is that the C. elegans worm first crossed over the ‘3 mm’ distance approaching the cuboid but then turned away from the cuboid (lower photos).

- Revised manuscript [Page 15, line 345 - 357]

We observed a particular behavior in individual C. elegans worms when they closely approached the side of the cuboid without post-curing and then rapidly turned away from the cuboid (< 14 sec) (Fig 4D, upper photos). This 'turn-away' motion could be attributed to formation of a gradient of toxic compounds released from the cuboid. We frequently observed this 'turn-away' motion within a 3 mm distance from the edge of the cuboid. Another observation we made was that some C. elegans worms initially crossed over the 3 mm distance while approaching the cuboid, but then turned away from the cuboid (lower photos). In contrast to the turning-away behavior observed in the uncured side, C. elegans on the cured control side displayed a different response. The worm approached and moved closer to the cured side without displaying the rapid turning-away motion seen in the uncured side. While occasional turns might still occur on the cured side, they were not as frequent or distinct as the ones observed on the uncured side.

9. For Fig 4E, authors should also provide the raw numbers used to calculate the percentage, so that readers may know how many worms are used to assess the trend. Also, method of statistical analysis should be mentioned.

<Answer>

Thank you for your comment. We would like to provide a supplementary table to show the raw numbers of worms counted in Fig. 4E. We have revised our manuscript to address the reviewer's comment and avoid any potential confusion or misunderstanding for the reader. Additionally, we used Student's t-test for the statistical analysis in our study [2].

 Distance from surface of the object 1st 2nd 3rd Average

Unmasked

(cured) 3mm 11 12 29 17.3

 6mm 18 19 63 33.3

 10mm 45 38 59 47.3

 Total 74 69 151 98

Masked

(uncured) 3mm 4 5 14 7.7

 6mm 10 19 27 18.7

 10mm 40 46 54 46.7

 Total 54 70 95 73

S1 Table 1. C. elegans distribution in each area.

 [The Authors’ response continues]

As supplementary table is provided, we revised our manuscript as following.

- Original manuscript [Page 15, line 360 - Page 16, line 362]

We found that C. elegans worms were evenly distributed regardless of the Al foil masked or unmasked regions with a distance range from zero to 10 mm (Fig 4E, left bars).

- Revised manuscript [Page 16, line 363 - 365]

We found that C. elegans worms were evenly distributed regardless of the Al foil masked or unmasked regions with a distance range from zero to 10 mm (Fig 4E, left bars, S1 Table).

10. The blue and yellow arrows in Fig 4E are not necessary.

<Answer>

Thank you. We realized that the blue and yellow arrows in Fig 4E may interfere with the reader's understanding. Therefore, we have taken your suggestion and deleted those two arrows.

- Original Fig 4E

- Revised Fig 4E

Reviewer #2: In this manuscript, Kim et al. present C. elegans, an animal model system, for toxicity evaluation using sensing and locomotion abilities for three-dimensional (3D) printing(3DP) extracts. The authors used C. elegans as an animal model to study the effects of 3DP extracts and observed that these extracts led to a reduction in lifespan, stress-gene expression, reproduction, and motility of the nematodes. The manuscript shows potential interest; however, some points need clarification:

Major Concerns:

1. The author needs to specify the choice of the used FUdR, as it is known to affect the differential expression of many genes.

<Answer>

Thank you for your valuable advice. It is true that hermaphroditic nematodes typically lay 200-300 eggs, which can be challenging to separate from an age-synchronized adult population upon hatching [3]. To address this issue, using FUDR in the experiments to confirm C. elegans longevity is indeed a viable option [4]. As you pointed out, FUDR can affect gene expression and has been reported to increase lifespan in some mutants. However, negligible effect of FUDR on the lifespan of wild-type has been reported [5–8]. 

The authors added the following sentences for discussion.

- Revised manuscript [Page 9, line 204 - 205]

~3D-printed cuboid was 15.8% (p<0.05). FUdR has been known to affect the differential expression of various genes. However, negligible effects of FUdR on the lifespan of wild-type has been reported. 

2. In online 143, the author mentions that "five L4/young adult stage worms were transferred to a fresh NGM plate containing 3D-printed block extract and permitted to lay eggs for 5hr." However, L4 stage worms are known to produce eggs after at least 8 hours. The authors should explain the egg-laying time.

<Answer>

Thank you for your comments. Wild-type C. elegans typically start laying eggs shortly after the L4/adult molt [9]. For our experiment, we used nematodes in the late L4 or early young adult stages, as egg-laying is not common during the early L4 stage. Within this time frame, the nematodes were able to lay a sufficient number of eggs within 8 hours. In some cases reported that egg-laying occurred within 4 hours [10–12]. We have revised the manuscript to provide more precision and clarity, in order to avoid misleading the readers.

- Original manuscript [Page 6, line 143 - Page 7, line 146]

For the fertility assay, five L4/young adult stage worms were transferred to a fresh NGM plate containing 3D-printed block extract and permitted to lay eggs for 5 hrs. The eggs were maintained at 20 °C for 2 days. A single worm was transferred to a fresh NGM plate containing 3D-printed block extract every day until it laid no eggs.

- Revised manuscript [Page 6, line 143 - 146]

For the fertility assay, five late L4 or early young adult stage worms were transferred to a fresh NGM plates seeded with contaminated OP50 and permitted to lay eggs for 5 hrs. The eggs were maintained at 20 °C for 2 days. A single worm was transferred to a fresh NGM plates seeded with contaminated OP50 every day until it laid no eggs.

3. The authors measured the number of eggs after the treatment of 3D-printed object extracts. Instead of showing produced daily eggs, the authors should show total progeny.

<Answer>

We appreciate your suggestion and agree that assessing the total progeny produced by the worms is a more appropriate measure of fertility compared to daily egg production. Fertility experiments typically focus on the ability of worms to produce viable offspring, while fecundity experiments are often used to assess egg hatching-related effects [13]. 

In response to your recommendation, we have conducted further analysis and provided Supplementary Figure 1 to demonstrate the total progeny. This figure shows the number of offspring produced by the worms as a measure of fertility, giving a more comprehensive understanding of the impact of 3D-printed object extracts on C. elegans reproduction. We have revised our manuscript accordingly to include S1 Fig. 

S1 Fig. Total number of progeny. The control group produced an average of 196 offspring, while C. elegans exposed to 3D-printed (3DP) extracts produced 138 offspring on average. Error bars represent standard error.

In light of the supplementary figure provided, we have revised the manuscript as follows.

- Original manuscript [Page 12, line 265 - 269]

The overall trend regarding the number of offspring per day showed a decreased number of offspring from the C. elegans exposed to the toxic compound LB extract: 8 at 2 days, 34 at 3 days, 11 at 4 days, and 4 at 5 days. Asterisks indicate p-values less than 0.05 compared with the control.

- Revised manuscript [Page 11, line 260 - Page 12, line 268]

The overall trend regarding the number of offspring per day showed a decreased number of offspring from the C. elegans exposed to the toxic compound LB extract: 8 at 2 days, 34 at 3 days, 11 at 4 days, and 4 at 5 days. Asterisks indicate p-values less than 0.05 compared with the control. Assessing the total progeny produced by the worms is a more appropriate measure of fertility compared to daily egg production. Fertility experiments typically focus on the ability of worms to produce viable offspring, while fecundity experiments are often used to assess simply egg hatching-related effects. We observed a decreasing trend in the total number of progeny upon exposure of 3D cuboid extracts (S1 Fig).

4. The authors should demonstrate the treatment's effect on the development stage and adult lethality for toxic effects in C. elegans."

<Answer>

We appreciate your suggestion to investigate the toxic effects on the developmental stages and adult lethality of C. elegans. We selected 2-methyl-4’-(methylthio)-2-morpholinopropiophenone as our test compound, as it is widely used toxic component of UV-curable 3D printing resins. In our experiments we allowed five L4/young adult stage nematodes to lay eggs and subsequently removed them to monitor egg hatching. Our results demonstrated that abnormal development began at the L3 stage and became more pronounced at the L4 stage, with smaller nematode sizes (red arrows) and unhealthy body shapes (blue arrow) in the treated group compared to the untreated one. 

We also investigated concentration-dependent toxicity behavior and found that overall behavioral toxicity was barely observed at a 0.5 mM concentration, while toxicity onset occurred at 5 mM. Although the 5 mM concentration might seem high, it is important to consider that the local concentration of toxic compounds leaching out from a particular defect point in 3D printed solid materials would be high. In contrast, the effective concentration affecting toxicity in a homogeneously distributed photoinitiator solution would be different due to the greater distance between photoinitiators. 

The authors have added these results to the supplementary data (S2 Fig) and incorporated relevant discussion points in our manuscript to address your concerns.

S2 Fig. Effects of photoinitiators on early stage development of C. elegans. “5 mM-M” represents 5 mM of 2-methyl-4′-(methylthio)-2-morpholinopropiophenone, and “0.5 mM-M” represents 0.5 mM of it. After L3 stage, “5 mM-M” showed abnormality of development. There were no significant differences observed between control and “0.5 mM-M” groups. Red bars indicate size of 1 mm.

[The Authors’ response continues]

The authors added the results above to supplementary data (S2 Fig) and several sentences of discussion. 

- Revised manuscript [Page 16, line 376 – Page 17, line 388]

~locomotion and sensing abilities. Furthermore, our study on the toxic effects of 2-methyl-4′-(methylthio)-2-morpholinopropiophenone, a widely used photoinitiator in 3D printing resins, on the developmental stage of C. elegans. The results showed that abnormal development began at the L3 stage, becoming more pronounced at the L4 stage, with smaller nematode sizes (red arrows) and unhealthy body shape (blue arrow) (S2 Fig). The concentration-dependent toxicity behavior showed that overall behavioral toxicity was barely observed at a 0.5 mM concentration, while toxicity onset occurred at 5 mM. This finding emphasizes the importance of considering local concentration of toxic compounds leaching out from a particular defect point of 3D printed solid materials when assessing toxicity as the effective concentration affecting toxicity in a homogeneously distributed solution may differ. This further highlights the utility of C. elegans as a unique model organism for detecting toxic compounds due to its combination of locomotion and sensing abilities.

References

1. Byerly L, Cassada R, Russell R. The life cycle of the nematode Caenorhabditis elegans: I. Wild-type growth and reproduction. Dev Biol. 1976;51: 23–33. 

2. Student. The probable error of a mean. Biometrika. 1908;6: 1–25. 

3. Rooney JP, Luz AL, González-Hunt CP, Bodhicharla R, Ryde IT, Anbalagan C, et al. Effects of 5′-fluoro-2-deoxyuridine on mitochondrial biology in Caenorhabditis elegans. Exp Gerontol. 2014;56: 69–76. doi:10.1016/j.exger.2014.03.021

4. Mitchell DH, Stiles JW, Santelli J, Sanadi DR. Synchronous Growth and Aging of Caenorhabditis elegans in the Presence of Fluorodeoxyuridine. J Gerontol. 1979;34: 28–36. doi:10.1093/geronj/34.1.28

5. Gandhi S, Santelli J, Mitchell DH, Wesley Stiles J, Rao Sanadi D. A simple method for maintaining large, aging populations of Caenorhabditis elegans. Mech Ageing Dev. 1980;12: 137–150. doi:10.1016/0047-6374(80)90090-1

6. Hosono R, Mitsui Y, Sato Y, Aizawa S, Miwa J. Life span of the wild and mutant nematode Caenorhabditis elegans. Exp Gerontol. 1982;17: 163–172. doi:10.1016/0531-5565(82)90052-3

7. Van Raamsdonk JM, Hekimi S. FUdR causes a twofold increase in the lifespan of the mitochondrial mutant gas-1. Mech Ageing Dev. 2011;132: 519–521. doi:10.1016/j.mad.2011.08.006

8. Aitlhadj L, Stürzenbaum SR. The use of FUdR can cause prolonged longevity in mutant nematodes. Mech Ageing Dev. 2010;131: 364–365. doi:10.1016/j.mad.2010.03.002

9. Schafer WR. Genetics of Egg-Laying in Worms. Annu Rev Genet. 2006;0. doi:10.1146/annurev.ge.40.111606.200001

10. Saha S, Guillily MD, Ferree A, Lanceta J, Chan D, Ghosh J, et al. LRRK2 Modulates Vulnerability to Mitochondrial Dysfunction in Caenorhabditis elegans. J Neurosci. 2009;29: 9210–9218. doi:10.1523/JNEUROSCI.2281-09.2009

11. Greer EL, Dowlatshahi D, Banko MR, Villen J, Hoang K, Blanchard D, et al. An AMPK-FOXO Pathway Mediates Longevity Induced by a Novel Method of Dietary Restriction in C. elegans. Curr Biol. 2007;17: 1646–1656. doi:10.1016/j.cub.2007.08.047

12. Park S-K, Tedesco PM, Johnson TE. Oxidative stress and longevity in Caenorhabditis elegans as mediated by SKN-1: SKN-1 response to oxidative stress. Aging Cell. 2009;8: 258–269. doi:10.1111/j.1474-9726.2009.00473.x

13. Rea SL, Ventura N, Johnson TE. Relationship Between Mitochondrial Electron Transport Chain Dysfunction, Development, and Life Extension in Caenorhabditis elegans. Kirkwood TBL, editor. PLoS Biol. 2007;5: e259. doi:10.1371/journal.pbio.0050259

---

## [Decision Letter · Decision Letter 1]

9 May 2023

PONE-D-23-04286R1Sniffer worm, C. elegans, as a toxicity evaluation model organism with sensing and locomotion abilitiesPLOS ONE

Dear Dr. Lee,

Thank you for submitting your manuscript to PLOS ONE. After careful consideration, we feel that it has merit but does not fully meet PLOS ONE’s publication criteria as it currently stands. Therefore, we invite you to submit a revised version of the manuscript that addresses the points raised during the review process.

As the reviewer 1 pointed out, the authors need to revise "Results and Discussion" section. The current version is not fit for publication. 

We look forward to receiving your revised manuscript.

Kind regards,

Myon-Hee Lee, Ph.D

Academic Editor

PLOS ONE

Reviewers' comments:

Reviewer's Responses to Questions

**Comments to the Author**

1. If the authors have adequately addressed your comments raised in a previous round of review and you feel that this manuscript is now acceptable for publication, you may indicate that here to bypass the “Comments to the Author” section, enter your conflict of interest statement in the “Confidential to Editor” section, and submit your "Accept" recommendation.

Reviewer #1: (No Response)

Reviewer #2: All comments have been addressed

2. Is the manuscript technically sound, and do the data support the conclusions?

Reviewer #1: Yes

Reviewer #2: Yes

3. Has the statistical analysis been performed appropriately and rigorously? 

Reviewer #1: Yes

Reviewer #2: Yes

4. Have the authors made all data underlying the findings in their manuscript fully available?

Reviewer #1: Yes

Reviewer #2: No

5. Is the manuscript presented in an intelligible fashion and written in standard English?

Reviewer #1: No

Reviewer #2: Yes

6. Review Comments to the Author

Reviewer #1: The authors have addressed all comments, except for the writing. The revised abstract and introduction is good, figure legends are good, but the 'Results and Discussion' section and "Conclusion" section is mostly left as it was before. In the current state, it is very distracting and difficult to read due to the poor writing, and it is unfit for publication.

The authors must significantly improve the writing to be considered for publication. I will point out a few examples that require attention. I have selected these examples, because these will not be addressed if the authors merely go through some minimal grammar edit, which will not be sufficient.

1. (p. 16, line 376) In the last part of the Results section, there's a part about worm development in 2-methyl-4'-(methylthio)-2-morpholinopropiophenone that was added in the revised manuscript. It's just tacked on in an odd location where it doesn't belong. The section deals with chemotaxis behavior and all of a sudden this data follows, without any transition or explanation. This part should be under its own subheading, maybe after the XPS data where the photoinitiator compound is first mentioned.

2. (p. 9, line 195) "Fig 1B clearly shows.."  the whole sentence structure will have to be fixed, but what I want to point out is, "clearly" is not a good way to describe a result. Whether it is clear or not, the reader will determine. What should be emphasized is whether it is statistically significant, which is a more objective metric.

3. (p. 11, line 250) "We designed an interesting experiment.."  again, the readers will determine whether it's interesting or not. Also, these types of fecundity tests are common in the literature, so not sure why you would comment that it's interesting.

4. (p.13, line 287) "This simple replacement results in a large difference in the toxicity mode of action."  why is this a large difference? It's different, but is it a "large" difference? The adjective is unnecessary and meaningless, and it inflates the significance of the experiment. If it is large, then it should be followed by more information explaining why it is large.

5. (p.13, line 289) "As C. elegans is able to swim by bending its body in an aqueous solution"  Because this sentence follows the previous sentence mentioning adverse substance being in water, it sounds like the aqueous solution the worm will swim in is the water extract. The sentence starts with "As", which suggests it's a consequence of something, but when you first read it, it's unclear whether the consequence is what's mentioned before the sentence ("but rather in water") or in the later part of the sentence ("we hypothesized that the speed of..").

These are just some examples. I think these reflect the basic expectations of how a research paper should be written, and should be easily fixed if someone more experienced can read over the manuscript.

I am especially troubled by the fact that the extra experiment added in the revised manuscript was carelessly tacked on at the end, in the same paragraph with an unrelated experiment. It is most unprofessional and it confounds me that the authors would go through the trouble of carrying out an extra experiment, but put so little effort in the writing.

Reviewer #2: This revised manuscript describes an important, general, and convincing finding of a toxicity evaluation model organism with sensing and locomotion abilities. I see no need for further revisions.

7. PLOS authors have the option to publish the peer review history of their article (what does this mean?). If published, this will include your full peer review and any attached files.

Reviewer #1: No

Reviewer #2: No

---

## [Author Response · Author response to Decision Letter 1]

29 May 2023

Reviewer #1: The authors have addressed all comments, except for the writing. The revised abstract and introduction is good, figure legends are good, but the 'Results and Discussion' section and "Conclusion" section is mostly left as it was before. In the current state, it is very distracting and difficult to read due to the poor writing, and it is unfit for publication.

The authors must significantly improve the writing to be considered for publication. I will point out a few examples that require attention. I have selected these examples, because these will not be addressed if the authors merely go through some minimal grammar edit, which will not be sufficient.

1. (p. 16, line 376) In the last part of the Results section, there's a part about worm development in 2-methyl-4'-(methylthio)-2-morpholinopropiophenone that was added in the revised manuscript. It's just tacked on in an odd location where it doesn't belong. The section deals with chemotaxis behavior and all of a sudden this data follows, without any transition or explanation. This part should be under its own subheading, maybe after the XPS data where the photoinitiator compound is first mentioned.

<Answer>

Thank you for your guidance. The reviewer’s opinion is correct. We re-located the corresponding paragraph from page 16, line 376-388 to page 14, line 324-339 as you recommended.

- Revised manuscript [Page 14, line 324 - Page 15, line 339]

In fact, XPS analysis showed that the nitrogen 1s signal was detected at 400.2 eV (Fig 3B), indicating that the observed toxicity of the extracted solution might originate from nitrogen-containing photoinitiators. Furthermore, our study on the toxic effects of 2-methyl-4′-(methylthio)-2-morpholinopropiophenone, a widely used photoinitiator in 3D printing resins, on the developmental stage of C. elegans. The results showed that abnormal development began at the L3 stage, becoming more pronounced at the L4 stage, with smaller nematode sizes (red arrows) and unhealthy body shape (blue arrow) (S2 Fig). The concentration-dependent toxicity behavior showed that overall behavioral toxicity was barely observed at a 0.5 mM concentration, while toxicity onset occurred at 5 mM. This finding emphasizes the importance of considering local concentration of toxic compounds leaching out from a particular defect point of 3D printed solid materials when assessing toxicity as the effective concentration affecting toxicity in a homogeneously distributed solution may differ. This further highlights the utility of C. elegans as a unique model organism for detecting toxic compounds due to its combination of locomotion and sensing abilities.

2. (p. 9, line 195) "Fig 1B clearly shows.."  the whole sentence structure will have to be fixed, but what I want to point out is, "clearly" is not a good way to describe a result. Whether it is clear or not, the reader will determine. What should be emphasized is whether it is statistically significant, which is a more objective metric.

<Answer>

Thank you for your valuable advice. We have revised the sentence to enhance its objectivity, incorporating your suggestion.

- Original manuscript [Page 9, line 195 - line 197]

Fig 1B clearly shows that the maximal lifespan of C. elegans living with the supplement of E. coli, which is grown in LB containing toxic compounds extracted from the cuboids, was decreased from 24 (black) to 20 (red) days.

- Revised manuscript [Page 9, line 195 - line 198]

Differences in the maximal lifespan of C. elegans were observed in Fig 1B, where the inclusion of E. coli supplemented with toxic compounds extracted from the cuboids in LB led to a notable reduction from 24 (black) to 20 (red) days.

3. (p. 11, line 250) "We designed an interesting experiment.."  again, the readers will determine whether it's interesting or not. Also, these types of fecundity tests are common in the literature, so not sure why you would comment that it's interesting.

<Answer>

Thank you for your valuable advice. We understand your point that the determination of whether an experiment is interesting or not should be left to the readers. Therefore, we have revised the sentence to remove the subjective language.

- Original manuscript [Page 11, line 250 - line 251]

We designed an interesting experiment to observe these toxic effects influencing the fertility of C. elegans.

- Revised manuscript [Page 11, line 252 - line 253]

We designed the experiment to observe these toxic effects influencing the fertility of C. elegans.

4. (p.13, line 287) "This simple replacement results in a large difference in the toxicity mode of action."  why is this a large difference? It's different, but is it a "large" difference? The adjective is unnecessary and meaningless, and it inflates the significance of the experiment. If it is large, then it should be followed by more information explaining why it is large.

<Answer>

Thank you for your valuable advice. We have taken your advice and eliminated the unnecessary and exaggerated wording from the sentence, resulting in a more accurate representation of the results.

- Original manuscript [Page 13, line 287 - line 288]

This simple replacement results in a large difference in the toxicity mode of action. The adverse substances are not in the food (i.e., E. coli) but rather in water.

- Revised manuscript [Page 13, line 289 - line 290]

The change in the extraction method would result in a difference in the toxicity mode of action. 

5. (p.13, line 289) "As C. elegans is able to swim by bending its body in an aqueous solution"  Because this sentence follows the previous sentence mentioning adverse substance being in water, it sounds like the aqueous solution the worm will swim in is the water extract. The sentence starts with "As", which suggests it's a consequence of something, but when you first read it, it's unclear whether the consequence is what's mentioned before the sentence ("but rather in water") or in the later part of the sentence ("we hypothesized that the speed of.."). 

<Answer>

Thank you for your valuable advice. We understand your point about the potential confusion caused by the sentence structure and its connection to the preceding statement. To clarify the sequence of events and hypotheses, we have revised the sentence accordingly.

- Original manuscript [Page 13, line 287 - line 294]

This simple replacement results in a large difference in the toxicity mode of action. The adverse substances are not in the food (i.e., E. coli) but rather in water. As C. elegans is able to swim by bending its body in an aqueous solution, we hypothesized that the speed of bending mobility could be affected. Approximately ten nematodes (age synchronized as 3 days) were transferred to either regular NGM or contaminated NGM by cuboid-extracted water (2 days). Finally, the worms on NGM were washed with M9 buffer and transferred to a 96-well plate to observe the bending speed of each group of C. elegans.

- Revised manuscript [Page 13, line 289 - line 299]

The change in the extraction method would result in a difference in the toxicity mode of action. Considering the presence of adverse substances in water not in the food (i.e., E. coli), we hypothesized that the speed of C. elegans’ bending mobility could be affected. To test this, we selected approximately ten nematodes, all of which were age-synchronized to 3 days. They were then placed in either standard NGM or contaminated NGM by using the cuboid-extracted water, which was prepared for 2 days. C. elegans is known to exhibit unique swimming behavior characterized by body bending in an aqueous environment. Therefore, any potential toxicity from the contaminated water might impact this behavior. After the exposure period, the nematodes were removed from the NGM, rinsed with M9 buffer, and transferred to a 96-well plate. This setup allowed us to analyze the bending speeds of the two groups of C. elegans. 

These are just some examples. I think these reflect the basic expectations of how a research paper should be written, and should be easily fixed if someone more experienced can read over the manuscript.

I am especially troubled by the fact that the extra experiment added in the revised manuscript was carelessly tacked on at the end, in the same paragraph with an unrelated experiment. It is most unprofessional and it confounds me that the authors would go through the trouble of carrying out an extra experiment, but put so little effort in the writing.

<Answer>

Thank you for your comments. The authors went through extensive grammar edits for the rest of paragraphs in results and discussion part as well as conclusion.

Relative expression levels of a stress-response gene

- Original manuscript [Page 10, line 226 - Page 11, line 243]

As we observed a lifespan reduction of C. elegans exposed to the extract of a 3D-printed cuboid, we assumed that there might be a gene indicator to provide further evidence at the molecular level. We chose hsp-16.2 and sod-3. hsp-16.2 is a stress-responsive reporter gene that predicts longevity in C. elegans [22]. The expression level of hsp-16.2 is correlated with increased resistance to heat-shock stress and increased lifespan in C. elegans [29]. sod-3 is an antioxidant gene involved in the cellular enzymatic defense system against reactive oxygen species [30]. The worms were exposed to 3D-printed cuboids for 2 days after the adult stage. We observed a significant downregulation of hsp-16.2 in worms treated with 3D-printed cuboid extract (Fig 1C upper panels and 1D left bars). Green fluorescent images were obtained because of the expression of the fusion protein HSP-16.2-GFP (Fig 1C). Qualitative analysis shows that the overall level of green fluorescence was considerably decreased. In addition, quantitative analysis (n = 20 for each group) was also performed. The relative expression level of hsp-16.2 decreased to 52.96 ± 6.37% for the C. elegans supplemented with the 3D-printed cuboid extract (Fig 1D, left bars). A similar result was obtained for SOD-3-GFP expression (74.92 ± 6.34% for the 3D cuboid extract group) (Fig 1C lower panels and 1D right bars). Therefore, the nematodes exposed to 3D-printed cuboid extract are thought to be less able to resist stress.

- Revised manuscript [Page 10, line 227 – Page 11, line 245]

As we observed a lifespan reduction of C. elegans exposed to the extract of a 3D-printed cuboid, we assumed that there might be a gene indicator to provide further evidences at a molecular level. We selected hsp-16.2 and sod-3. Hsp-16.2, a stress-responsive reporter gene, has been shown to predict longevity in C. elegans, and its expression correlates with increased resistance to heat-shock stress and enhanced lifespan [22, 29]. Sod-3, on the other hand, plays a role in the cellular enzymatic defense system against reactive oxygen species [30]. To investigate these genes, we exposed the worms to 3D-printed cuboids for 2 days post-adulthood. We observed a significant downregulation of hsp-16.2 in worms treated with 3D-printed cuboid extract (Fig 1C upper panels and 1D left bars). The observation was based on green fluorescent images obtained due to the expression of the fusion protein HSP-16.2-GFP (Fig 1C). The qualitative analysis showed a considerable decrease in overall green fluorescence. In addition, quantitative analysis (n = 20 for each group) was also performed. The relative expression level of hsp-16.2 decreased to 52.96 ± 6.37% for the group supplemented with the 3D-printed cuboid extract (Fig 1D, left bars). A comparable result was seen with SOD-3-GFP expression (74.92 ± 6.34% for the 3D cuboid extract group) (Fig 1C lower panels and 1D right bars). These findings suggest that the nematodes exposed to 3D-printed cuboid extract have diminished stress resistance capabilities, reinforcing our previous findings regarding the reduction in lifespan.

C. elegans is a unique animal model system with sensing and locomotion abilities 

- Original manuscript [Page 14, line 331 - Page 17, line 388]

As mentioned in the introduction section, C. elegans is able to move away from any positions from which harmful toxic compounds are locally present or released. In contrast, its locomotion direction is not affected in nontoxic environments [15]. The chemotactic properties shown by C. elegans can point out a source position from which toxic compounds are released. To this point, the experiments described in Figs 1-3 demonstrated that photoinitiators and other toxic compounds were released from the 3D cuboids up to a certain toxicity level. Thus, we designed an interesting experiment when fabricating a 3D cuboid. First, a cuboid with the same dimensions (1 x 1 x 0.1 cm3) was printed and then was illuminated with UV light for post-curing to minimize the release of toxic compounds. Second, when the post-curing step was carried out, a piece of Al foil was wrapped around one-half of the side of the cuboid (Fig 4A). We hypothesized that nearly all toxic compounds would be released from the half block without post-UV curing. In fact, the cuboid without post-curing was slightly bent upon applying external force (Fig 4B).

We observed a particular behavior in individual C. elegans worms when they closely approached the side of the cuboid without post-curing and then rapidly turned away from the cuboid (< 14 sec) (Fig 4D, upper photos). This 'turn-away' motion could be attributed to formation of a gradient of toxic compounds released from the cuboid. We frequently observed this 'turn-away' motion within a 3 mm distance from the edge of the cuboid. Another observation we made was that some C. elegans worms initially crossed over the 3 mm distance while approaching the cuboid, but then turned away from the cuboid (lower photos). In contrast to the turning-away behavior observed in the uncured side, C. elegans on the cured control side displayed a different response. The worm approached and moved closer to the cured side without displaying the rapid turning-away motion seen in the uncured side. While occasional turns might still occur on the cured side, they were not as frequent or distinct as the ones observed on the uncured side.

In addition to the aforementioned description of the dynamic motion of an individual C. elegans, we observed the overall population behavior in a collection. We observed geometrical distributions of all C. elegans organisms centered on the cuboid at 1 hr after worm introduction at the designated four points (Fig 4C). The reason for the 1 hr setting time is to establish the release and gradient formation of toxic substances from the cuboids. We found that C. elegans worms were evenly distributed regardless of the Al foil masked or unmasked regions with a distance range from zero to 10 mm (Fig 4E, left bars, S1 Table). In other words, nearly 50% of the population of C. elegans was found in the masked region, and the remaining 50% was found in the unmasked region. However, when we looked closely at the region from zero to 6 mm, differences in geometrical distributions were found. Approximately 61% of the C. elegans worm were crawling in the photocured region, and 39% of the population was found in the masked region (the middle bars) (p-value = 0.0531). This difference in chemotactic behavior was further enhanced when we observed the crawling worms living from zero to 3 mm from the cuboid. Approximately 70% of the C. elegans worms were crawling in the photocured region, and 30% of the population was found in the masked region (the right bars) (p-value < 0.001). The results clearly indicate that C. elegans can be a unique model organism for detecting toxic compounds because of the combination of locomotion and sensing abilities. Furthermore, our study on the toxic effects of 2-methyl-4′-(methylthio)-2-morpholinopropiophenone, a widely used photoinitiator in 3D printing resins, on the developmental stage of C. elegans. The results showed that abnormal development began at the L3 stage, becoming more pronounced at the L4 stage, with smaller nematode sizes (red arrows) and unhealthy body shape (blue arrow) (S2 Fig). The concentration-dependent toxicity behavior showed that overall behavioral toxicity was barely observed at a 0.5 mM concentration, while toxicity onset occurred at 5 mM. This finding emphasizes the importance of considering local concentration of toxic compounds leaching out from a particular defect point of 3D printed solid materials when assessing toxicity as the effective concentration affecting toxicity in a homogeneously distributed solution may differ. This further highlights the utility of C. elegans as a unique model organism for detecting toxic compounds due to its combination of locomotion and sensing abilities.

- Revised manuscript [Page 15, line 351 - Page 17, line 394]

As mentioned in the introduction section, C. elegans is able to move away from positions where harmful toxic compounds are locally present or being released, while its locomotion direction remains unaffected in non-toxic environments [15]. The chemotactic properties shown by C. elegans can help identify the source of toxic compound release. This is particularly relevant in light of the experiments described in Figs 1-3, which demonstrated that photoinitiators and other toxic compounds were released from the 3D cuboids up to a certain toxicity level. Thus, we designed an experiment when fabricating a 3D cuboid. A cuboid with identical dimensions (1 x 1 x 0.1 cm3) was printed, followed by UV light illumination for post-curing to minimize the release of toxic compounds. Simultaneously, during the post-curing step, we wrapped a piece of Al foil around one half of the cuboid (Fig 4A). Our hypothesis was that majority of toxic compounds would be released from the half block without post-UV curing. In fact, the non-post-cured cuboid exhibited slight bending upon applying external force (Fig 4B).

We observed distinct behavioral patterns in individual C. elegans worms when they closely approached the non-post-cured side of the cuboid, rapidly turning away from the cuboid within less than 14 sec. (Fig 4D, upper photos). This 'turn-away' motion may be attributed to the formation of a gradient of toxic compounds released from the cuboid. We frequently observed this 'turn-away' action within a 3 mm distance from the edge of the cuboid. It is noteworthy that some C. elegans worms initially crossed over the 3 mm distance while approaching the cuboid, but then quickly turned away from it (lower photos). However, in contrast to the turning-away behavior observed on the non-post-cured side, C. elegans approaching the cured side (control) showed a different response. They moved closer to the cured side without displaying the rapid turning-away motion observed on the non-cured side. While occasional turns might still occur on the cured side, they were not as frequent as the ones observed on the uncured side.

Alongside these observations of the individual C. elegans, we also examined the overall population behavior. We monitored the geometrical distributions of all C. elegans organisms centered on the cuboid, one hour after worm introduction at the designated four points (Fig 4C). We opted for the 1 hr setting time to allow the release and gradient formation of toxic substances from the cuboids. We found that C. elegans worms were evenly distributed across both the Al foil masked or unmasked regions within a distance range from zero to 10 mm (Fig 4E, left bars, S1 Table). In other words, approximately 50% of the population of C. elegans was found in the masked region, with the remaining 50% located in the unmasked region. Upon further examination of the region from zero to 6 mm, however, we noticed differences in the geometrical distributions. Approximately 61% of the C. elegans were found in the photocured region, with 39% located in the masked region (the middle bars) (p-value = 0.0531). This difference in chemotactic behavior became more pronounced when we observed the crawling worms within a range from zero to 3 mm from the cuboid. Approximately 70% of the C. elegans worms were located in the photocured region, and 30% in the masked region (the right bars) (p-value < 0.001). The results indicate that C. elegans can serve as a unique model organism for detecting toxic compounds because of its combination of locomotion and sensing abilities.

Conclusion 

- Original manuscript [Page 17, line 404 - Page 18, line 416]

Typical model organisms that have been used in the past have contributed a large amount to toxicity assessments of substances. However, typical model organisms are not suitable for determining in which direction chemicals diffuse from leachable materials. We chose 3D-printed products as typical leachable materials. By using a small nematode, C. elegans, approximately 1 mm in body length, we were able to carry out a toxicity test that would be difficult to perform with conventional model organisms. The 3D-printed block extract adversely affected the health of C. elegans, resulting in reduced lifespan, reduced exercise capacity, reduced number of offspring, and decreased relative expression levels of stress-response genes. In addition, the leaching point and the direction of chemical diffusion could be monitored by following the movement of worms located near the substances. Material toxicity assessments that could not have been conducted with conventional model organisms can be performed by tracking the distribution and behavior of C. elegans.

- Revised manuscript [Page 18, line 410 - line 426]

Conventional model organisms that have greatly contributed to toxicity assessments of substances in the past. However, these typical model organisms do not offer an adequate solution for discerning the direction in which chemicals diffuse from leachable materials. In this study, we selected 3D-printed products as representative leachable materials and used a small nematode, C. elegans, which is approximately 1 mm in body length, for our toxicity test. This approach allowed us to conduct a toxicity test that would be challenging to perform with traditional model organisms. Our results showed that the extract from the 3D-printed block adversely affected the health of C. elegans, leading to a shortened lifespan, reduced physical activity, diminished offspring count, and decreased relative expression levels of stress-response genes. Furthermore, by tracking the movement of the worms in proximity to the substance, we were able to monitor the leaching point and the direction of chemical diffusion of toxic components. This research demonstrates that material toxicity assessments, which could not have been conducted with conventional model organisms, can now be performed by monitoring the distribution and behavior of C. elegans. As such, C. elegans proves itself to be a valuable model organism, offering insight into the understanding and assessment of material toxicity in a dynamic manner.

---

## [Decision Letter · Decision Letter 2]

12 Jun 2023

PONE-D-23-04286R2Sniffer worm, C. elegans, as a toxicity evaluation model organism with sensing and locomotion abilitiesPLOS ONE

Dear Dr. Lee,

Thank you for submitting your manuscript to PLOS ONE. After careful consideration, we feel that it has merit but does not fully meet PLOS ONE’s publication criteria as it currently stands. Therefore, we invite you to submit a revised version of the manuscript that addresses the points raised during the review process.

As you can see the reviewer's comment, your revised version is not sufficient for publication because of a poor writing. ==============================

We look forward to receiving your revised manuscript.

Kind regards,

Myon-Hee Lee, Ph.D

Academic Editor

PLOS ONE

Reviewers' comments:

Reviewer's Responses to Questions

**Comments to the Author**

1. If the authors have adequately addressed your comments raised in a previous round of review and you feel that this manuscript is now acceptable for publication, you may indicate that here to bypass the “Comments to the Author” section, enter your conflict of interest statement in the “Confidential to Editor” section, and submit your "Accept" recommendation.

Reviewer #1: (No Response)

2. Is the manuscript technically sound, and do the data support the conclusions?

Reviewer #1: Yes

3. Has the statistical analysis been performed appropriately and rigorously? 

Reviewer #1: Yes

4. Have the authors made all data underlying the findings in their manuscript fully available?

Reviewer #1: Yes

5. Is the manuscript presented in an intelligible fashion and written in standard English?

Reviewer #1: No

6. Review Comments to the Author

Reviewer #1: Despite the two rounds of revisions, the writing is still very poor. The edits made are insufficient to bring it up to the level fit for publication.

7. PLOS authors have the option to publish the peer review history of their article (what does this mean?). If published, this will include your full peer review and any attached files.

Reviewer #1: No

---

## [Author Response · Author response to Decision Letter 2]

14 Jul 2023

6. Review Comments to the Author

Reviewer #1: Despite the two rounds of revisions, the writing is still very poor. The edits made are insufficient to bring it up to the level fit for publication.

Thank you very much for your comments. 

The authors carried out comprehensive editing and revision throughout the manuscript keeping efforts not to change the original meaning of each sentence. 

Abstract

Three-dimensional (3D) printed objects have a higher likelihood of containing local defects compared to products manufactured by conventional casting-based methods.

-> Additive manufacturing, or 3D printing, has revolutionized the way we create objects. However, its layer-by-layer process may lead to an increased incidence of local defects compared to traditional casting-based methods.

The multistep, layer-by-layer process in additive manufacturing, with factors such as light intensity, variations of light penetration depth, inhomogeneity of components, and fluctuation of nozzle temperature, contribute to the increased defect generations.

-> Factors such as light intensity, depth of light penetration, component inhomogeneity, and fluctuations in nozzle temperature all contribute to defect formations.

Defect regions are sources of toxic component release, but methods to identify them in printed materials have not been reported.

-> These defective regions can become sources of toxic component leakage, but pinpointing their locations in 3D printed materials remains a challenge.

Traditional toxicological assays rely on extraction followed by treatments to living creatures, which is a passive detection of harmful agents.

-> Traditional toxicological assessments rely on the extraction and subsequent exposure of living organisms to these harmful agents, thus only offering a passive detection approach.

Thus, developing of an active system to identify and locate the sources of toxicity is a critical issue in 3D printing technologies.

-> Therefore, the development of an active system to both identify and locate sources of toxicity is essential in the realm of 3D printing technologies.

Herein, we present the use of the nematode model system, C. elegans, for toxicity evaluation.

-> Herein, we introduce the use of the nematode model organism, Caenorhabditis elegans (C. elegans), for toxicity evaluation.

C. elegans displays unique ‘sensing’ and ‘locomotion’ abilities, actively moving toward safe regions while avoiding toxic areas.

-> C. elegans exhibits distinctive 'sensing' and 'locomotion' capabilities that enable it to actively navigate toward safe zones while steering clear of hazardous areas.

These capabilities make C. elegans an unparalleled choice among existing aquatic and animal models, offering immediate and precise indications to locate and identify the sources of toxicity in 3D printed materials.

-> This active behavior sets C. elegans apart from other aquatic and animal models, making it an exceptional choice for immediate and precise identification and localization of toxicity sources in 3D printed materials.

Introduction

The assessment necessitates the use of living organisms to determine material toxicity

-> Assessment of material toxicity fundamentally requires testing on living organisms.

This plant’s small size, easy of cultivation, and short life cycle make it a practical choice for toxicity evaluations [3,4]

-> Lemna gibba's small size, simple cultivation process, and short life cycle render it a practical candidate for toxicity assessments [3,4]

Primary advantage of using this plant lies in the ease of observing growth inhibition by toxic components.

-> The primary advantage of utilizing this plant model resides in the observation of growth inhibition induced by toxic elements.

However, the results obtained from Lemna gibba growth inhibition did not directly correspond to toxicological effects on humans.

-> However, the outcomes obtained from Lemna gibba growth inhibition studies do not directly parallel the toxicological impacts on humans.

Due to the limitations associated with the correlation between plant-based toxicity models and human toxicity levels, later studies have introduced animal models.

-> Due to the discrepancies between plant-based toxicity models and human toxicity outcomes, subsequent research has embraced the use of animal models.

However, the results obtained in aquatic environments are very different from those in ambient conditions, leading to a preference for terrestrial animal models.

-> However, the findings from aquatic settings vary significantly from those observed in terrestrial conditions, thus inclining researchers towards the use of terrestrial animal models.

However, implementation of stringent and wide-ranging ethical regulations has recently posed a significant challenge to the use of these model systems [10].

-> Nonetheless, the recent enforcement of rigorous and comprehensive ethical regulations has introduced considerable challenges to the employment of these model systems [10].

Reasons include that layer-by-layer deposition processes in additive manufacturing essentially involves numerous repeated steps, during which unavoidable defects are generated.

-> This is primarily due to the intrinsic layer-by-layer deposition processes in additive manufacturing, which inherently involve multiple iterations, during which the generation of defects is inescapable.

In stereolithography (SLA) methods, variations in light intensity, penetration depth, and inhomogeneity of polymers concentrations and dispersion can occur [13].

-> In stereolithography (SLA) methods, variances in factors such as light intensity, penetration depth, and the concentration and dispersion inhomogeneity of polymers can arise [13].

Furthermore, the irradiation with UV-light leads to a homolytic bondage cleavage and the generation of highly reactive radical species, which are the origin of chemical toxicity.

-> Furthermore, UV-light irradiation triggers a homolytic bond cleavage, resulting in the creation of highly reactive radical species, which serve as the roots of chemical toxicity.

In fused deposition modeling (FDM), nozzle temperature fluctuations contribute to the formation of local defects during printing [14].

-> In fused deposition modeling (FDM), fluctuations in nozzle temperature significantly contribute to the development of local defects during the printing process [14].

As 3D printing techniques are ideal for small-scale prototype production, each product may display unique sites of defects, resulting in varying levels of toxicity.

-> Given that 3D printing techniques are optimally suited for small-scale prototype production, each product may exhibit distinctive defect sites, thereby leading to disparate levels of toxicity.

Thus, there is a need to develop a new animal model capable of detecting product-to-product variations in toxicity sources.

-> Consequently, it becomes imperative to develop a novel animal model capable of detecting variations in toxicity sources between individual products.

Plant models lack the locomotive function to identify toxicological sources within printed materials.

-> Plant models, lacking locomotive capabilities, fail to discern toxicological sources within printed materials.

While zebrafish, an aquatic toxicity evaluation model, has a locomotive function, leached toxic compounds become diffused in water, making them unsuitable for pinpointing toxicity sources in printed materials.

-> Although zebrafish, an aquatic model used in toxicity evaluations, possess locomotive abilities, the diffusion of leached toxic compounds in water impedes their effectiveness in accurately locating toxicity sources within printed materials.

In light of these challenges, a new toxicity model should combine both sensing and locomotion capabilities.

-> Given these challenges, it becomes critical to introduce a toxicity model that unifies both sensing and locomotion capabilities.

Here, we propose the use of C. elegans as a model capable of detecting both the overall toxicity and specific toxic sites within printed materials.

-> We propose the employment of Caenorhabditis elegans (C. elegans), a model endowed with the capacity to discern both the overall toxicity and specific toxic sites within printed materials.

The nematode’s crawling activity serves as phenotypic indicators for identifying the sources of toxicity within test materials.

-> The crawling activity of the nematode serves as phenotypic indicators pivotal for pinpointing the sources of toxicity within test materials.

However, the potential of C. elegans as an ‘active’ toxicological model, incorporating both sensing and locomotion abilities, has not been explored previously.

-> Yet, the potential of C. elegans as an 'active' toxicological model, integrating both sensing and locomotion capabilities, has been unexplored. 

Results and Discussion

- Original manuscript [Page 8, line 189 – Page 9, line 201]

Effect of 3D printed blocks extracted LB on lifespan

We hypothesized the following. First, Luria-Bertani broth (LB) media were prepared because the energy source of C. elegans is bacteria, for which Escherichia coli (OP50) has been widely used. Second, any toxic compounds released from 3D-printed cuboids (1 x 1 x 0.1 cm3, 80 units incubation in 400 mL of LB, Fig 1A) might be adsorbed during E. coli growth. Third, the C. elegans eating the contaminated E. coli might exhibit a reduced lifespan. Differences in the maximal lifespan of C. elegans were observed in Fig 1B, where the inclusion of E. coli supplemented with toxic compounds extracted from the cuboids in LB led to a notable reduction from 24 (black) to 20 (red) days. A control experiment used LB media without toxic compound extraction from the cuboids. Importantly, the mean lifespan, defined by 50% survival of C. elegans, was 15 days for the control, which was decreased to approximately 13 days for the treatment group (dotted lines, Fig 1B).

- Revised manuscript [Page 9, line 194 – line 205]

Toxic substance effects on C. elegans’ lifespan

Release of toxic compounds released from the 3D-printed cuboids was monitored by observing the lifespan of C.elegans. Fig 1B shows the differences in the maximal lifespan of C. elegans. When C.elegans was exposed to the toxic compounds through the Escherichia. Coli (OP50) supplemented with toxic compounds extracted from the cuboids in Luria-Bertani broth (LB) media, the lifespan was reduced from 24 (black) to 20 (red) days. We hypothesized that toxic compounds released from the 3D-printed cuboids are absorbed by OP50 during the growth in LB media. A control experiment was carried out with the OP50 grown in LB media without the toxic compounds extracted from the cuboids. The mean lifespan, defined by 50% survival of C. elegans, was 15 days for the control, which was decreased to 13 days for the treatment group (dotted lines, Fig 1B).

- Original manuscript [Page 10, line 227 - line 245]

As we observed a lifespan reduction of C. elegans exposed to the extract of a 3D-printed cuboid, we assumed that there might be a gene indicator to provide further evidences at a molecular level. We selected hsp-16.2 and sod-3. Hsp-16.2, a stress-responsive reporter gene, has been shown to predict longevity in C. elegans, and its expression correlates with increased resistance to heat-shock stress and enhanced lifespan [22, 29]. Sod-3, on the other hand, plays a role in the cellular enzymatic defense system against reactive oxygen species [30]. To investigate these genes, we exposed the worms to 3D-printed cuboids for 2 days post-adulthood. We observed a significant downregulation of hsp-16.2 in worms treated with 3D-printed cuboid extract (Fig 1C upper panels and 1D left bars). The observation was based on green fluorescent images obtained due to the expression of the fusion protein HSP-16.2-GFP (Fig 1C). The qualitative analysis showed a considerable decrease in overall green fluorescence. In addition, quantitative analysis (n = 20 for each group) was also performed. The relative expression level of hsp-16.2 decreased to 52.96 ± 6.37% for the group supplemented with the 3D-printed cuboid extract (Fig 1D, left bars). A comparable result was seen with SOD-3-GFP expression (74.92 ± 6.34% for the 3D cuboid extract group) (Fig 1C lower panels and 1D right bars). These findings suggest that the nematodes exposed to 3D-printed cuboid extract have diminished stress resistance capabilities, reinforcing our previous findings regarding the reduction in lifespan.

- Revised manuscript [Page 10, line 230 – Page 11, line 248]

The lifespan reduction of C. elegans exposed to the extract of a 3D-printed cuboid indicates that there might be a gene indicator to provide further evidences at a molecular level. We selected hsp-16.2 and sod-3. Hsp-16.2, a stress-responsive reporter gene, has been known to predict longevity in C. elegans, and its expression correlates with increased resistance to heat-shock stress and enhanced lifespan [22, 29]. Sod-3, on the other hand, plays a role in the cellular enzymatic defense system against reactive oxygen species [30]. When the worms were exposed to the 3D-printed cuboid extracts for 2 days post-adulthood, we observed a significant downregulation of hsp-16.2 in the worms treated with 3D-printed cuboid extract (Fig 1C upper panels and 1D left bars). The observation was made with the green fluorescent images due to the expression of the fusion protein HSP-16.2-GFP (Fig 1C). The qualitative analysis showed a considerable decrease in overall green fluorescence. In addition, quantitative analysis (n = 20 for each group) was also performed. The relative expression level of hsp-16.2 decreased to 52.96 ± 6.37% for the group supplemented with the 3D-printed cuboid extracts (Fig 1D, left bars). A comparable result was obtained with SOD-3-GFP expression (74.92 ± 6.34% for the 3D cuboid extract group) (Fig 1C lower panels and 1D right bars). These findings suggest that the nematodes exposed to 3D-printed cuboid extracts have diminished the stress resistance capabilities, supporting our previous findings regarding the reduction in lifespan. 

- Original manuscript [Page 11, line 247 - line 253]

We assume that the toxic substances released from the cuboids might affect the fertility of C. elegans. In general, approximately 300 eggs are hatched in the entire lifetime of a healthy nematode [31]. Thus, tens of eggs can be found in a plate even after one day of culture of a single C. elegans organism [32]. Negative effects on fertility can be observed for C. elegans organisms that are exposed to the toxic substances released into LB. We designed the experiment to observe these toxic effects influencing the fertility of C. elegans. First, all C. elegans nematodes were synchronized in their age.

- Revised manuscript [Page 11, line 245 – line 251]

We thought that the toxic substances released from the cuboids could impact the fertility of C. elegans. Normally, healthy nematode can hatch approximately 300 eggs throughout its lifetime [31], resulting in the presence of tens of eggs even after one day of culturing a single C. elegans organism [32]. Exposure to the toxic substances in LB medium can adversely affect the fertility of C. elegans. Thus, we designed an experiment to investigate the influence of these toxic effects on their fertility. Firstly, all C. elegans nematodes were synchronized in their age.

- Original manuscript [Page 12, line 285 – Page 13, line 309]

For the previous LB case, we observed overall negative toxicity results regarding lifespan, stress-response genes, and fertility. These results are due to the contaminated E. coli bacteria that were prepared in media into which toxic substances had been released. A different type of extraction is replacing LB media with distilled water (Fig 2C). The change in the extraction method would result in a difference in the toxicity mode of action. Considering the presence of adverse substances in water not in the food (i.e., E. coli), we hypothesized that the speed of C. elegans’ bending mobility could be affected. To test this, we selected approximately ten nematodes, all of which were age-synchronized to 3 days. They were then placed in either standard NGM or contaminated NGM by using the cuboid-extracted water, which was prepared for 2 days. C. elegans is known to exhibit unique swimming behavior characterized by body bending in an aqueous environment. Therefore, any potential toxicity from the contaminated water might impact this behavior. After the exposure period, the nematodes were removed from the NGM, rinsed with M9 buffer, and transferred to a 96-well plate. This setup allowed us to analyze the bending speeds of the two groups of C. elegans. We counted the number of body bends for 1 minute. Fig 2E shows time-lapse photo arrays representing body bending. For a healthy C. elegans worm, it takes approximately 1.6 sec to bend its body to an opposite position (upper photos). However, we observed that twice the time was required for the same bending motion in the case of the exposed worms (lower photos). Quantitative analysis showed that the unexposed worms (control) bent 53.2 times per minute (n = 6), which was decreased to 24 bends per minute for the exposed C. elegans (n = 6) (Fig 2D). Although we cannot exclude the possibility of the OP50 taking up the toxins from the NGM media, these results clearly demonstrate the adverse effect of contaminated water produced from cuboid extraction on C. elegans motility.

- Revised manuscript [Page 13, line 286 – Page 14, line 310]

The overall negative effects on lifespan, stress-response genes, and fertility were attributed to the toxic substances released into the media which contaminated E. coli bacteria. To investigate a different mode of toxicity effect, we employed a different extraction method by replacing LB media with distilled water (Fig 2C). This change in the extraction method provides a difference in the toxicity mode of action. The presence of toxic substances in water rather than in the food (i.e., E. coli) may induce the change of C. elegans’ bending mobility. To test this, we selected approximately ten nematodes, all of which were age-synchronized to 3 days. They were then placed in either standard NGM or contaminated NGM by using the cuboid-extracted water. C. elegans is known to exhibit unique swimming behavior characterized by body bending in an aqueous environment. Therefore, any potential toxicity in the contaminated water might affects this behavior. After the exposure, the nematodes were removed from the NGM, rinsed with M9 buffer, and transferred to a 96-well plate. This setup allowed us to analyze the bending speeds of the two groups of C. elegans. We counted the number of body bends for 1 minute. Fig 2E shows time-lapse photo arrays representing body bending. For a healthy C. elegans worm, it takes approximately 1.6 sec to bend its body to an opposite position (upper photos). However, we observed that twice the time was required for the same bending motion in the case of the exposed worms (lower photos). Quantitative analysis showed that the unexposed worms (control) bent 53.2 times per minute (n = 6), which was decreased to 24 bends per minute for the exposed C. elegans (n = 6) (Fig 2D). Although we cannot exclude the possibility of the OP50 taking up the toxins from the NGM media, these results show the adverse effect of contaminated water with the cuboid extracts on C. elegans motility.

- Original manuscript [Page 14, line 313 - line 329]

In general, biological toxicity when using photocurable resins is largely dependent on the photoinitiators [33]. However, knowledge of the resin composition used in this study is quite limited due to the company’s proprietary right. Nevertheless, it is expected that the resin contains at least a certain amount of photoinitiators. We thought that unreacted photoinitiators would remain inside of the 3D-printed products until they eventually leached out and adversely affected the health of C. elegans. In Fig 3A, the chemical structures of widely used UV-curable photoinitiators are presented: 2-benzyl-2-(dimethylamino)-4’-morpholinobutyrophenone, 4-(dimethylamino)benzophenone, azobisisobutyronitrile, and 2-methyl-4-2-morpholinopropiophenone. Nitrogen atoms commonly exist because they undergo homolytic cleavage upon introducing light energy [34]. For this reason, we assumed that the extracted solution might have nitrogen-containing compounds. In fact, XPS analysis showed that the nitrogen 1s signal was detected at 400.2 eV (Fig 3B), indicating that the observed toxicity of the extracted solution might originate from nitrogen-containing photoinitiators. Furthermore, our study on the toxic effects of 2-methyl-4′-(methylthio)-2-morpholinopropiophenone, a widely used photoinitiator in 3D printing resins, on the developmental stage of C. elegans.

- Revised manuscript [Page 14, line 314 – line 329]

Biological toxicity of the photocurable resins comes from the photoinitiators [33]. However, the detailed information of the resin composition used in this study is quite limited due to the company’s proprietary right. Nevertheless, we thought that the resin contains certain amount of photoinitiators. We hypothesized that unreacted photoinitiators would remain within the 3D-printed products until they leach out over time, potentially causing adverse effect on the health of C. elegans. Fig 3A illustrates the chemical structures of widely used UV-curable photoinitiators: 2-benzyl-2-(dimethylamino)-4’-morpholinobutyrophenone, 4-(dimethylamino)benzophenone, azobisisobutyronitrile, and 2-methyl-4-2-morpholinopropiophenone. These compounds have nitrogen atoms, which undergo homolytic cleavage upon exposure to light energy [34]. Consequently, the extracted solution contains the nitrogen-containing compounds. In fact, XPS analysis revealed that the nitrogen 1s signal was detected at 400.2 eV (Fig 3B), indicating that the observed toxicity of the extracted solution might originate from the nitrogen-containing photoinitiators. Further, we investigated the toxic effects of 2-methyl-4′-(methylthio)-2-morpholinopropiophenone, a widely used photoinitiator in 3D printing resins, on the developmental stage of C. elegans.

- Original manuscript [Page 15, line 351 – Page 16, line 364]

As mentioned in the introduction section, C. elegans is able to move away from positions where harmful toxic compounds are locally present or being released, while its locomotion direction remains unaffected in non-toxic environments [15]. The chemotactic properties shown by C. elegans can help identify the source of toxic compound release. This is particularly relevant in light of the experiments described in Figs 1-3, which demonstrated that photoinitiators and other toxic compounds were released from the 3D cuboids up to a certain toxicity level. Thus, we designed an experiment when fabricating a 3D cuboid. A cuboid with identical dimensions (1 x 1 x 0.1 cm3) was printed, followed by UV light illumination for post-curing to minimize the release of toxic compounds. Simultaneously, during the post-curing step, we wrapped a piece of Al foil around one half of the cuboid (Fig 4A). Our hypothesis was that majority of toxic compounds would be released from the half block without post-UV curing. In fact, the non-post-cured cuboid exhibited slight bending upon applying external force (Fig 4B).

- Revised manuscript [Page 15, line 351 – Page 16, line 364]

As mentioned in the introduction section, C. elegans possesses the ability to detect and move away from locations where harmful toxic compounds are present or being released, while maintaining its locomotion in non-toxic environments [15]. The chemotactic behavior displayed by C. elegans can be leveraged to identify the source of toxic compound release. In the context of the experiments described in Figs 1-3, which demonstrated that photoinitiators and other toxic compounds could be released from the 3D cuboids up to a certain toxicity level. Thus, we designed an experiment when fabricating a 3D cuboid. A cuboid with identical dimensions (1 x 1 x 0.1 cm3) was printed, followed by UV light illumination for post-curing to minimize the release of toxic compounds. Simultaneously, during the post-curing step, we wrapped a piece of Al foil around one half of the cuboid (Fig 4A). Our hypothesis was that majority of toxic compounds would be released from the half block without post-UV curing. Indeed, the non-post-cured cuboid exhibited slight bending upon applying external force (Fig 4B).

- Original manuscript [Page 18, line 410 - line 426]

Conventional model organisms that have greatly contributed to toxicity assessments of substances in the past. However, these typical model organisms do not offer an adequate solution for discerning the direction in which chemicals diffuse from leachable materials. In this study, we selected 3D-printed products as representative leachable materials and used a small nematode, C. elegans, which is approximately 1 mm in body length, for our toxicity test. This approach allowed us to conduct a toxicity test that would be challenging to perform with traditional model organisms. Our results showed that the extract from the 3D-printed block adversely affected the health of C. elegans, leading to a shortened lifespan, reduced physical activity, diminished offspring count, and decreased relative expression levels of stress-response genes. Furthermore, by tracking the movement of the worms in proximity to the substance, we were able to monitor the leaching point and the direction of chemical diffusion of toxic components. This research demonstrates that material toxicity assessments, which could not have been conducted with conventional model organisms, can now be performed by monitoring the distribution and behavior of C. elegans. As such, C. elegans proves itself to be a valuable model organism, offering insight into the understanding and assessment of material toxicity in a dynamic manner.

- Revised manuscript [Page 18, line 410 – line 426]

Conventional model organisms have played a significant role in past toxicity assessments of various substances. However, these typical models lack the capability to accurately determine the direction of chemical diffusion from leachable materials. In this study, we employed 3D-printed products as representative leachable materials and utilized the small nematode C. elegans, which measures approximately 1 mm in body length, for our toxicity test. This approach enabled us to conduct a toxicity assessment that would have been challenging with traditional model organisms. Our findings showed that the extract obtained from the 3D-printed block had detrimental effects on the health of C. elegans, resulting in a shortened lifespan, reduced physical activity, decreased offspring count, and diminished expression levels of stress-response genes. Moreover, by tracking the movement of the worms in the vicinity of the substance, we were able to observe the leaching point and ascertain the direction of chemical diffusion of the toxic components. This research demonstrates the utility of C. elegans as a model organism, providing insights into the understanding and evaluation of material toxicity in a dynamic manner. Consequently, material toxicity assessments that were previously impractical with conventional model organisms can now be performed by monitoring the distribution and behavior of C. elegans.

---

## [Decision Letter · Decision Letter 3]

20 Jul 2023

Sniffer worm, C. elegans, as a toxicity evaluation model organism with sensing and locomotion abilities

PONE-D-23-04286R3

Dear Dr. Lee,

We’re pleased to inform you that your manuscript has been judged scientifically suitable for publication and will be formally accepted for publication once it meets all outstanding technical requirements.

The reviewer also asked you to correct a few of minor typos during a final proofread.

line 232,234: gene names are in lowercase, even if they are in the beginning of a sentence.

line 247: "extracts have diminished the stress resistance capabilities"  delete "the"

line 267: "Asterisks indicate p values less than 0.05 compared with the control"  usually these things are indicated in the figure legend, not in the manuscript text itself.

line 298: "affects"  "affect" 

Kind regards,

Myon-Hee Lee, Ph.D

Academic Editor

PLOS ONE

Additional Editor Comments (optional):

Reviewers' comments:

Reviewer's Responses to Questions

**Comments to the Author**

1. If the authors have adequately addressed your comments raised in a previous round of review and you feel that this manuscript is now acceptable for publication, you may indicate that here to bypass the “Comments to the Author” section, enter your conflict of interest statement in the “Confidential to Editor” section, and submit your "Accept" recommendation.

Reviewer #1: All comments have been addressed

2. Is the manuscript technically sound, and do the data support the conclusions?

Reviewer #1: (No Response)

3. Has the statistical analysis been performed appropriately and rigorously? 

Reviewer #1: (No Response)

4. Have the authors made all data underlying the findings in their manuscript fully available?

Reviewer #1: (No Response)

5. Is the manuscript presented in an intelligible fashion and written in standard English?

Reviewer #1: (No Response)

6. Review Comments to the Author

Reviewer #1: The writing has improved considerably. I recommend the manuscript for publication.

Just a few minor typos:

line 232,234: gene names are in lowercase, even if they are in the beginning of a sentence.

line 247: "extracts have diminished the stress resistance capabilities"  delete "the"

line 267: "Asterisks indicate p values less than 0.05 compared with the control"  usually these things are indicated in the figure legend, not in the manuscript text itself.

line 298: "affects"  "affect"

7. PLOS authors have the option to publish the peer review history of their article (what does this mean?). If published, this will include your full peer review and any attached files.

Reviewer #1: No

---

## [Editor Report · Acceptance letter]

24 Jul 2023

PONE-D-23-04286R3 

Sniffer worm, *C. elegans*, as a toxicity evaluation model organism with sensing and locomotion abilities 

Dear Dr. Lee:

I'm pleased to inform you that your manuscript has been deemed suitable for publication in PLOS ONE. Congratulations! Your manuscript is now with our production department. 

Kind regards, 

on behalf of

Dr. Myon-Hee Lee 

Academic Editor

PLOS ONE